# How many cyberbullying(s)? A non-unitary perspective for offensive online behaviours

**Stefano Guidi**[ID]*, **Paola Palmitesta**[ID], **Margherita Bracci, Enrica Marchigiani, Ileana Di Pomponio, Oronzo Parlangeli**

Department of Social, Political and Cognitive Sciences, University of Siena, Siena, Italy

* stefano.guidi@unisi.it

**Data Availability Statement:** All relevant data are within the manuscript and its Supporting Information files (S2 File).

## Abstract

Research has usually considered cyberbullying as a unitary phenomenon. Thus, it has been neglected to explore whether the specific online aggressive behaviours relate differentially to demographic features of the perpetrators of online aggressive actions, their personality characteristics, or to the ways in which they interact with the Internet. To bridge this gap, a study was conducted through a questionnaire administered online to 1228 Italian high-school students (Female: 61.1%; 14–15 yo: 48.%; 16–17 yo: 29.1%; 18–20 yo: 20.4%, 21–25 yo: 1.6%; Northern Italy: 4.1%; Central Italy: 59.2%; Southern Italy: 36.4%). The questionnaire, in addition to items about the use of social media, mechanisms of Moral Disengagement and personality characteristics of the participants in the study, also included a scale for the measurement of cyberbullying through the reference to six aggressive behaviours. The results indicate that cyberbullying can be considered as a non-unitary phenomenon in which the different aggressive behaviours can be related to different individual characteristics such as gender, personality traits and the different ways of interacting with social media. Moreover, the existence of two components of cyberbullying has been highlighted, one related to virtual offensive actions directly aimed at a victim, the other to indirect actions, more likely conducted involving bystanders. These findings open important perspectives for understanding, preventing, and mitigating cyberbullying among adolescents.

## Introduction

Social media and the Internet are today the prevailing means of communication among adolescents and young adults. While online communication media provide many opportunities, their use also exposes individuals to the risk of being victims of aggressive behaviours and misconduct. A single action such as posting an embarrassing photo or video of someone on a social network is enough for this to be seen by many people many times [1–4], with potential negative consequences for the victim, such as low self-esteem, distress, depression, loneliness, sadness [5–8]. Such negative consequences may be even worrisome. A meta-analysis of 47 studies found that bullying victimization and bullying perpetration may be associated with suicidal ideation and behaviour [9].

**Funding:** The author(s) received no specific funding for this work.

**Competing interests:** The authors have declared that no competing interests exist.

A very recent review on cyberbullying among adolescents and children, based on 63 references [10], reported that the prevalence of cyberbullying perpetration varied across studies from 6.0% to 46.3%, while the rates of cyberbullying victimization from 14.0% to 57.5%. Other authors [3, 11–13] underline that about 20–40% of adolescents are harassed online. These numbers have been constantly on the rise [14, 15], and the number of studies aimed at analysing cyberbullying has increased as well. In a recent analysis, it was highlighted that since 2011/2012 there has been a linear increase in scientific articles on cyberbullying each year, reaching 268 publications in 2019 [15].

In this regard, several reviews have already been produced [3, 4, 16–18] that showed how research has essentially focused on some fundamental aspects related to cyberbullying, and that there are still many open issues.

First, we can note some difficulties in the definition of the phenomenon [19], though some common characteristics of cyberbullying behaviours have been highlighted so far. In generally accepted definitions, cyberbullying has several key characteristics: it is a behaviour intended to hurt, it is enacted by one or more people, it is performed through an electronic medium, and it is directed at those who cannot easily defend themselves [2]. According to [20], cyberbullying should be distinguished from cyber incivility and cyber aggression. The former one, in fact, does not involve an imbalance of power, and should refer to any rude or rough behaviour carried out through electronic means. Cyber aggression, on the other hand, refers to those behaviours aimed at hurting, causing injury, and like cyber victimization does not involve an imbalance of power [21]. Cyberbullying thus appears to be based essentially on an *abuse of power* [22, 23] that is manifested by behaviours that are systematically aimed at causing harm and that are carried out using electronic means [2].

However, definitions can be different depending on one's point of view, and acting like a bully can mean different things to different people. For example, in a study conducted on a UK sample, O'Brien [24] showed that teachers tend to qualify as bullying a higher number of behaviours, on the basis of factors such as the way the behaviour affects the victim, gender discrimination, besides power relations between the aggressor and the victim.

Above all, there are two aspects in the attempts to define the peculiarities of cyberbullying that are rather opaque, as pointed out by Slonje et al. [4]. The first has to do with the issue of *repetition of behaviour*. The Internet, in fact, can be a powerful amplifier and booster of any word or action. And so even the one-time offense can become "endless." The other aspect has to do with determining what might contribute to an imbalance of power between aggressor and victim. As argued by Vandebosch & van Cleemput [25], this may also relate to greater technological experience or to the ability to behave anonymously. Moreover, a meta-analysis of 22 studies, has shown that often the role of the victim and the role of the bully can involve the same person, and the occurrence of this condition may depend on the culture of reference, i.e. Central European, Mediterranean, North American, South American and Asian cultures [26].

It seems interesting to highlight that, recently, several studies have been trying to refine algorithms with the aim of detecting offensive behaviour in different social media [27–29]. These studies are often based on the results provided by the psychological analysis of recent years, thus confirming the role of personality factors [28, 30]. They also contribute to better define what cyberbullying is and detect it more readily, for instance by capturing repetitive behaviour and sentiment information [31, 32]. To date, however, these studies are often based on datasets collected from a single or few social media [28, 30, 32, 33], and they do not always consider all the different offensive behaviours, such as those concerning ethnicity, religion, physical and cognitive impairments [30].

Because of these blurry aspects, it seems really difficult to clearly qualify what cyberbullying is and to distinguish it from other forms of virtual aggressions.

## Variables associated with cyberbullying

Even when we consider the subjective variables, the results are not always homogeneous. Above all, with respect to gender and age, the studies do not show consistent data, although the trend seems to indicate a greater harassment towards girls, with a peak for bullying behaviour between the ages of 12 and 15 [2, 34–36] (for a recent review of studies on the subject see [10]). Tokunaga [3] suggested a curvilinear trend with a peak that can be identified between 7th and 8th grades, while Barlett & Coyne [37], by directly analysing the relationships between gender and age, showed an interesting relationship between these two variables. Generally, it is boys who are more involved in offending behaviours. However, this difference between the sexes in the likelihood to engage in offensive behaviour seems to become evident only as boys get older.

Another key factor for predicting aggressive behaviour on the net is given by the time spent on it, which, if excessive, is often associated with psychological distress. A meta-analysis showed a correlation between a problematic use of the Internet and depression, though this effect is essentially associated with male gender [38] Moreover, some scholars found a correlation among aggressive virtual behaviour, a greater experience in the use of communication technologies and time spent online [39–42]. At the same time when social media—the communication means that are most involved in acts of cyberbullying—are taken into account, they do not seem to have a direct link to misconduct; though the use of social media that most easily allow behaviours in anonymity seems more connected to the occurrence of offensive behaviours [13]. However, intensive use of social media seems more clearly to be a risk factor for the victim more than for the perpetrator [39].

Several studies have also focused on the personality characteristics of bullies (see, for instance [43]). Among the most investigated aspects are those related to personality traits in reference to the theory of the big five [44]. In an analysis aimed at distinguishing bullying from cyberbullying [45], it was found that, unlike so-called traditional bullying, cyberbullying is essentially negatively related to the trait of agreeableness. This finding has emerged also in other studies in which, however, cyberbullying was also found to be negatively related to conscientiousness [42] and to affective empathy [46]. Recently, a study attempted to relate the personality characteristics of Twitter users to their offensive messages [47]. Using an automated categorization of both personality characteristics and user behaviours, it was highlighted that extraversion, agreeableness, and neuroticism are highly relevant predictors (up to 96% accurate) for identifying those Twitter users who can be qualified as bullies.

More specifically, personality characteristics have been related to Moral Disengagement, that is those thought mechanisms that are exercised to diminish the sanctioning aspects of moral principles [48–50]. In essence, moral disengagement mechanisms are justifications for one's thoughts or actions that are made acceptable by referring to moral principles that are considered at a higher level, such as when sacrificing an individual for the collective interest or punishing those who are thought to deserve punishment. It has been previously reported that subjects who self-report having engaged in cyberbullying differ from those who have never done so for a greater reliance on all moral disengagement mechanisms [13]. This finding had already emerged in a previous study by Meter & Bauman [51], who found that moral disengagement is related as much to cyberbullying as it is to traditional bullying. Again, however, the relationships between moral disengagement and cyberbullying are not fully defined. The use of moral disengagement mechanisms in the context of online relationships can be

particularly variable, and thus quite different from those involved in bullying perpetrated in non-virtual contexts. In fact, while traditional bullying seems to be context independent, the same appears not to be true for cyberbullying [52]. From the results of the analysis conducted by Paciello and colleagues [52], the possibility emerges that online moral disengagement should be formulated and studied as a theoretical construct separate from the one which manifests in traditional contexts, although the two constructs stand out as related.

## Measuring cyberbullying

Another issue related to the understanding, definition and prevention of cyberbullying refers directly to the instruments commonly used to measure its occurrence. According to many studies the different manifestations of bullying, such as cyberbullying, are better explained as indicators of a unidimensional construct [53–56]. In fact, the many instruments used and proposed in the literature generally involve the use of a variable number of items that, with ordinal, interval, or ratio response scales [57] aim to uniquely qualify the experience of cyberbullying. In 2014, an analysis conducted to evaluate self-report instruments in reference to cyberbullying [58] showed that many scales were already available. More precisely, 27 scales were identified, although not all of them at an adequate level of development and with satisfactory validity and reliability indices. Today it can be said that there are scales with a long tradition such as the revised version of the Olweus Bully-Victim Questionnaire [59], the Peer Relations Questionnaire [60], and the Forms of Bullying Scale [55]. And it was from these three scales that Thomas and colleagues [57] validated a scale that allowed them to assert that the construct of cyberbullying is essentially one-dimensional (see also [61]). Despite this result, however, assumptions can be made in relation to the fact that the results obtained are evidently determined by the behaviours included in the scale which may be more or less similar to each other.

In a recent review of tools used for measuring cyberbullying Chun et al. [62] highlighted that of the 64 studies considered, only 15 followed a correct approach to scale construction with respect to item selection. Furthermore, only 1 of these had subscales to assess perpetration-only cyberbullying. This is the 19 items scale proposed by Álvarez-García et al. [34] inspired directly by the multifactorial model proposed by Palladino et al. [63]. In these and a few other cases (see [64]), an attempt was made to structure scales that would account for the different behavioural facets of cyberbullying through an appropriate number of items. Factor analyses conducted on the data identified some factors (two to four) that could provide a finer understanding of the phenomenon. These studies, however, have not attempted to account for both the various subjective (e.g. personality factors) or contextual factors (e.g., the possibility to have frequent interactions on the Internet) that are likely to be involved in motivating the different manifestations of online aggressive behaviours.

Other scales, such as the one used by Meter & Bauman [51] were created with an intent that clearly aims at brevity. This tool presents only 6 items for behaviours that describe typical cyberbullying actions by different means, which are paralleled with as many items on cyber-victimization. Specifically, those misconducts have to do with verbal aggressiveness, sending nasty messages, revealing secrets, spreading gossip, visual aggressiveness, and impersonation. Usually these items are analysed in a unified manner, for example as the authors did in reference to moral disengagement [51]. However, an analysis that attempts to relate the different behaviours to different subject characteristics could be informative at a more fine-grained level. Therefore, it seems reasonable to try to analyse whether the different cyberbullying behaviours described within the same scale could be predicted by different subjective and contextual-behavioural variables. This could lead to a more clearly referable understanding of the phenomenon in its different manifestations.

## The study

The aim of the study was to investigate different forms of cyberbullying involving the use of social media and the internet by adolescents, to understand their determinants with respect to individual psychological variables and to explore associations with demographic variables and variables related to the pattern of social networks use.

Four different research questions (RQ) were addressed in the study:

1. *RQ1. Are there differences in the frequency of different types of cyberbullying behaviours*, *overall and as a function of socio-demographic variables*? We might expect that different forms of aggressive behaviour have different frequencies (H1) while it is not clear whether differences in cyberbullying frequency across categories such as gender (i.e. more frequently reported by boys) or age should be found [37] for all the different behaviours (H2) or not.

2. *RQ2. Are different types of cyberbullying behaviours differently related to individual features such as personality traits and moral disengagement*? It is known from several studies [13, 51, 52] that cyberbullying is more common in adolescents who exhibit certain characteristics such as a high level of moral disengagement, and some personality characteristics as neuroticism. Escortell and colleagues [65] found a relationship among online aggressors and a minor level of agreeableness and conscientiousness and major level of neuroticism, while it has been observed that agreeableness and openness to experience are [66, 67] a key factor to be a victim. Although there is no general consensus among the findings on this issue, we can suppose a relationship between *specific* forms of aggressive behaviour and personality, cognitive and behavioural factors (H3).

3. *RQ3. Are different types of cyberbullying behaviours specifically related to different profiles of internet and social media usage*? Previous research has shown that aggressive behaviour on the internet is related to the time spent online and using social media [68], but it is not clear whether this association can be found for cyberbullying in general or whether it is specific to certain forms of misconduct only (H4). The latter assumption can be made if we take a multidimensional perspective of cyberbullying. Moreover, with few exceptions [13], factors such as the number of accounts or followers on social networks have not been adequately studied in relation to cyberbullying.

4. *RQ4. Is cyberbullying better conceived and measured as a unitary construct or as a multidimensional construct*? It is possible that inconsistencies among findings about the association of aggressive behaviours online and demographic or individual variables might be due to differences in the measures of cyberbullying used in the studies. We hypothesise that even investigating a small number of behaviours, as in the case of Meter & Bauman's scale [51], non-unified perspectives of cyberbullying may emerge (H5).

To address these research questions, and test the various hypotheses described above, we structured a questionnaire that was administered to a large sample of high school students.

## Materials and methods

### Participants and procedure

One thousand two hundred and twenty-eight (1228) students from high-schools, technical and professional schools, participated in the study. The majority were girls (61.1%), and half of the participants were in the 14–15 age range (48.4%), followed by the 16–17 age range (28.7%) and by 18–20 years (20.2%). Only 19 students belonged to the 21–25 age range (1.5%). Demographic information about participants in the sample are presented in Table 1.

**Table 1. Socio-demographic characteristics of the sample.**

| *Variable* | *N* | *%* |
|---|---|---|
| Gender | *NA = 2* | |
| Female | 750 | 61.2% |
| Male | 476 | 38.8% |
| Age | *NA = 14* | |
| 14–15 years | 594 | 48.9% |
| 16–17 years | 353 | 29.1% |
| 18–20 years | 248 | 20.4% |
| 21–25 years | 19 | 1.6% |
| Geographical region [a] | | |
| Northern Italy | 50 | 4.1% |
| Central Italy | 723 | 58.9% |
| Southern Italy and Islands | 445 | 36.2% |

Descriptive statistics (frequencies) about the socio-demographic characteristics of the sample (N = 1.228).
[a]Regions in Northern Italy: Trentino Alto-Adige, Lombardia, Valle d'Aosta, Piemonte; Regions in Central Italy: Tuscany, Lazio; Regions in Southern Italy: Campania, Sicilia.

The schools were distributed across the Italian country (Northern Italy: 4.1%; Central Italy: 59.2%; Southern Italy: 36.4%) and were chosen via personal contacts of a small group of university students who served as research assistants in the study and that attended those schools before enrolling to college. The headmasters of the schools were contacted and informed about the study. The parents of minors were also contacted through the school and asked to give consent to the participation of their child to the study. After having received their authorization, students were asked to complete a self-reported on-line questionnaire on a voluntary basis. The ethical aspects of the study were approved by the department, acting as Ethical Committee, in March 2019 (report n. 10/2019 of 13 March 2019). The different schools involved were all contacted at different times during 2019, and data collection ended in December 2019.

## Materials and measures

The questionnaire included 32 questions, and was structured into 5 sections, dedicated to collect, respectively, socio-demographic information about participants, information on their patterns of social media use, their personality traits [69], the levels of moral disengagement about cyberbullying [51] and the frequency of cyberbullying behaviours [51].

In the first section participants were asked to report their gender, their age, the city in which they lived, and the type of secondary school they were attending.

The use of social networks was investigated in the second section through multiple choice questions on the most frequently used social media (Youtube, Instagram, Snapchat, Google+, Whatsapp, Twitter, Facebook, etc), and the time spent online every day ("never"; "less than 1 hour"; "1 to 3 hours"; "more than 3 hours"). Feelings related to network use—such as jitters if unable to use the Internet were also inquired ("If I can't use Internet I feel nervous") were measured on a five-point agreement scale (1 = strongly disagree / 5 = strongly agree). Also the number of social profiles were investigated ("None"; "One"; "More than one") as well as the number of followers of each participant ("Less than 200 followers", "More than 200 to 500", "More than 500 to 1000", "More than 1000 to 5000", "More than 5000 followers").

The last three sections of the questionnaire, as detailed in the following paragraphs, included the Italian version of three scales aimed at measuring personality traits [70], moral disengagement about cyberbullying and cyberbullying behaviours [50, 51] respectively.

**Big Five Inventory (BFI).** Personality was measured using the Italian version of the Big Five Inventory-10 [69, 70], a very brief scale comprising only 10 items, which measures the five personality traits from the Big Five model: Agreeableness, Extraversion, Conscientiousness, Emotional Stability, and Openness to Experience. Responses for all the items are expressed on a 5-point agreement scale (1 = strongly disagree and 5 = strongly agree), and each trait is measured by two items, with one item for each scale reverse coded. Despite its brevity, this instrument has shown good psychometrics properties in the validation study [70].

**Moral disengagement about cyberbullying.** Moral Disengagement about cyberbullying was measured using the Italian version of a self-report scale proposed by Meter and Bauman [51] and including eight items. Each item refers to a different type of cyberbullying behaviour, for example "Cyberbullying annoying classmates is just teaching them a lesson" or "It's okay to treat someone badly if they behave like a jerk", and responses are expressed on a 5-point agreement scale, where 1 corresponds to "strongly disagree" and 5 to "strongly agree". The scores are then averaged across items to get a summary measure of the level of moral disengagement. The items had been translated in Italian by the authors for a previous study [68] in which a confirmatory factor analysis had shown excellent fit and prediction validity.

**Cyberbullying.** Cyberbullying perpetration was measured using the Italian version of a self-report scale devised by Meter and Bauman [51], which includes 6 items, each referring to a different type of online aggressive behaviour directed against someone: 1) "Sending a mean or nasty text message"2) "Sending mean or nasty email messages"; 3) "Sending an embarrassing photo of someone via cell phone", 4) "Pretended to be someone else on the Internet"; 5) "Revealing someone else's secrets online or by cell phone without their permission"; 6) "Spreading a rumour about someone on the Internet". For each behaviour, participants were asked to report how often they had carried it out, on a 4-point frequency scale ranging from never to more than five times (1 = never; 2 = 1–2 times; 3 = 3–5 times; 4 = 5+ times). The items had been translated in Italian by the authors for a previous study [68] in which a confirmatory factor analysis had shown excellent fit.

The full questionnaire used in the study is available in S1 Text.

## Statistical analysis

Descriptive statistics (means and standard deviations for numeric variables, and frequency tables for categorical ones) were computed to characterise the composition of the sample with respect to socio-demographic features, personality characteristics of respondents, social network use, moral disengagement about cyberbullying and perpetration of cyberbullying acts. Chi-squared tests were used to compare frequency distributions across gender and age, and t-tests to compare the means of numerical variables across genders.

Pearson product-moment correlations between personality factors, aggressive behaviours and moral disengagement about cyberbullying were computed to understand the possible relationships between these variables. Holm's method was used to adjust p-values for multiple comparisons.

Ordinal logistic regression models were used to investigate predictors of the self-reported frequency of the different cyberbullying behaviours, measured on an ordinal scale with 4 levels ("never", "1 to 2 times", "3 to 5 times", "more than 5 times"). For each behaviour, a separate regression model was used, using ratings of the self-reported frequency of perpetration as dependent variable. Predictors included in all the models were: age, gender, the five personality

factors, the negative feelings when they do not have access to the internet, the time spent on social media, the number of social profiles and moral disengagement scores. All the numerical predictors were centred on the mean. For ordinal and nominal predictors were used treatment coding, and the reference categories for the categorical variables were chosen in the following way. Gender was coded so that it was 0 for "female", and 1 for "male". For age, the smaller age class was used as the reference category. For the number of followers and the time on social networks, which both had 3 levels, the middle category was chosen as the reference ("200–500 followers", "1–3 h/d on social networks"), while the number of social network accounts was coded as 0 for those with only one account, and 1 for those with more than one account. Internet addiction, which was measured on a 5-point scale, was treated as a numerical variable, and centred on the mean. All the predictors were entered simultaneously in the regression models. Participants who had reported to never use social networks, were excluded from the regression analyses.

To analyse the factorial structure of the measurement of cyberbullying we have used a combination of Exploratory and Confirmatory Factor Analysis, conducted respectively on 30% (train set) and 70% (test set) of the data (randomly split). EFA was conducted using minimal residual for factor extraction and oblimin rotation. Polychoric correlations were used for the analysis, and parallel analysis was used to determine the number of factors to retain. In the CFA we tested two models, the one extracted by the EFA and a single factor model. The indicators were left on the categorical, ordered, response scale, and diagonally weighted least squares (DWLS) was used to estimate the model parameters. The fitness of the models was compared using several fit indexes (Comparative Fit Index—CFI; Normed Fit Index—NFI; Non-Normed Fit Index—NNFI; Root Mean Square Error Approximation—RMSEA).

The measurement model that had better fit in the CFA was used in a structural equation modelling analysis (SEM). The effects of personality traits (measured by the Italian BFI-10 scores) on the two cyberbullying factors (latent dependent variables), direct and mediated by moral disengagement (latent mediator variable) were tested in the model, along with the direct effect of a measure on an ordinal scale of internet addiction. A multigroup SEM analysis was conducted for each of these grouping factors: gender, time on social networks and having (vs not having) more than one profile on social networks. Differences in the means of the latent variables across groups in each category were tested after assessing measurement invariance (configural, weak and strong), using the following analytical strategy. For each grouping factor, first we conducted a multigroup SEM analysis allowing all the parameters in the model to vary freely across groups (i.e. parameters were estimated separately for each group), and assessed the fit of the multigroup model using fit indexes to verify whether configural invariance held (good fit) or not. Weak and strong configural invariance were then tested fitting a second and a third multigroup model in which the loadings of the latent factors (step 2) *and* the intercept of the latent variables' indicators (step 3) were constrained to be equal across groups, allowing all the other parameters to vary, and comparing the significance of change in $\chi^2$. A non-significant change was taken as evidence that the more restrictive model (higher invariance) fitted data as well as the less restrictive model, and therefore should be preferred (being more parsimonious) [71].

Personality traits were included as observed variables (aggregated scores of the two items in each BFI trait), as indicated for this brief scale Moderation of the effects of personality traits and moral disengagement by each of the grouping factors was also assessed in the multigroup analysis, by estimating parameters for the change in the slope of each effect across groups.

All the analyses were conducted using R version 4.0.2 [72], using the functions in the package *lavaan* (version 0.6–9) [73]. All the code and data used in the analysis are available online in a compressed folder (S2 File). In Fig 1 a visual summary of the methodology is presented.

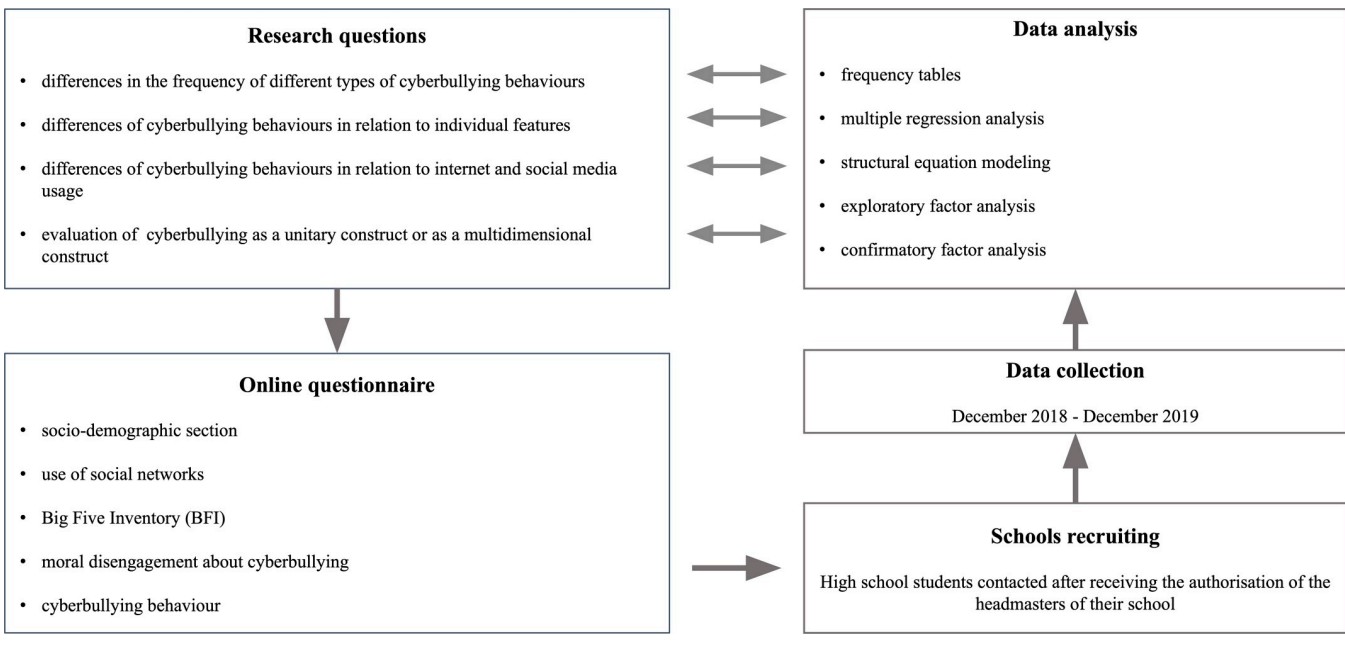

**Fig 1. Visual summary of the methodology.**

## Results

### Cyberbullying perpetration

*RQ1. Are there differences in the frequency of different types of cyberbullying behaviours, overall and as a function of socio-demographic variables?*

We computed the percentages of the responses concerning the self-reported frequency of perpetration of the six different cyberbullying behaviours included in the questionnaire. The percentages are plotted in Fig 2 as stacked bar charts. For each behaviour, most participants reported they had never perpetrated it, and only a minority (from 2% to 18%), reported having it done three or more times. A significant Chi-square test showed that the distribution of self-reported frequencies varied across cyberbullying behaviours ($\chi^2$(15) = 631.2, p < .001). The lowest frequency of perpetration was found for sending mean or indecent emails (never enacted by 94.1% of participants), and the highest for sending embarrassing photos (more than once by 48.1% of participants).

The distribution of the self-reported cyberbullying perpetration frequency varied significantly across genders for three behaviours: *sending indecent messages* ($\chi^2$(3) = 40.67, p < .001), *sending indecent emails* ($\chi^2$(3) = 8.92, p = .028), and *sending embarrassing photos* ($\chi^2$(3) = 10.06, p = .018). Girls were more likely to have never sent indecent messages (63.9%) than boys (48.2%), and less likely to have sent them more than five times (girls: 5.6%, boys: 13.1%). Girls were also more likely to have never sent indecent emails (95.5%) or embarrassing photos (58.6%) than boys (never sent indecent emails: 92.0%, never sent embarrassing photos: 54.9%).

### Personality and moral disengagement

*RQ2. Are different types of cyberbullying behaviours differently related to individual features such as personality traits and moral disengagement?*

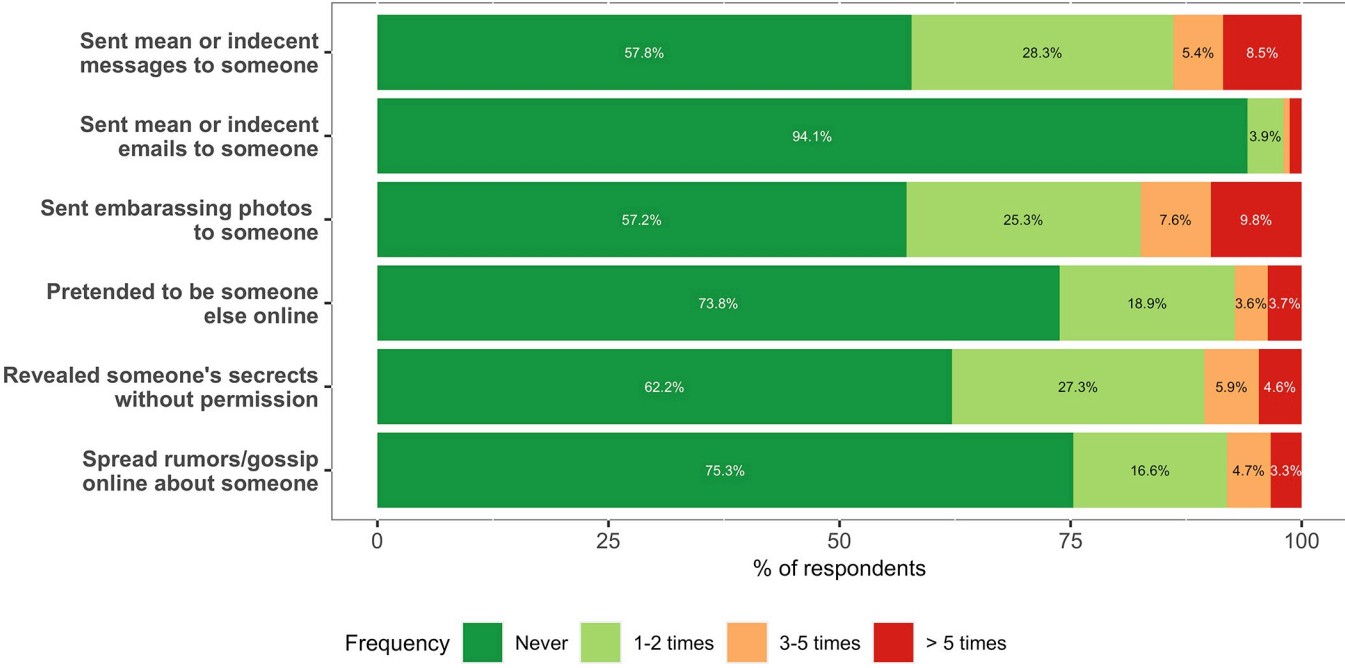

**Fig 2. Self-reported frequency of cyberbullying.** Stacked frequencies bar charts showing the distribution of responses about the self-reported frequency of different types of cyberbullying behaviours.

In Table 2 are reported the average scores for the measures of personality traits and moral disengagement in the sample, overall and by gender. A series of independent sample t-tests showed that girls were significantly less extravert and agreeable, they had less recourse to moral disengagement mechanisms and were less emotionally stable and open than boys. The average score on the question about feeling nervous without access to the internet is also reported in the bottom row (M = 2.69, SD = 1.18), and for this variable no significant differences were found by gender and age group.

**Table 2. Descriptive statistics about personality, moral disengagement and internet addiction.**

|  | All participants | Females | Males | p-value |
|---|---|---|---|---|
|  | *N = 1228* | *N = 750* | *N = 476* |  |
| Extraversion | 3.35 (0.93) | 3.25 (0.93) | 3.50 (0.90) | **<0.001** |
| Agreeableness | 3.10 (0.87) | 3.03 (0.86) | 3.22 (0.86) | **<0.001** |
| Conscientiousness | 3.27 (0.92) | 3.26 (0.92) | 3.30 (0.92) | 0.402 |
| Emotional Stability | 2.73 (1.06) | 2.49 (1.04) | 3.11 (0.98) | **<0.001** |
| Openness | 3.35 (1.00) | 3.40 (0.97) | 3.26 (1.03) | **0.024** |
| Moral Disengagement | 1.77 (0.63) | 1.64 (0.54) | 1.96 (0.71) | **<0.001** |
| Internet addiction | 2.69 (1.18) | 2.73 (1.18) | 2.64 (1.18) | 0.183 |

Average scores and standard deviations for the measures of the big five personality traits, the level of moral disengagement about cyberbullying and the ratings of the degree to which participants feel nervous (Internet addiction) when they do not have access to the internet (from 1 to 5). These descriptive statistics were computed both over the entire sample, collapsing across the gender of respondents, and separately for female and male participants. The p-values reported in the rightmost column are derived from independent samples t-test comparing the average scores for male and female participants for each measure.

## Use of social media/networks

*RQ3. Are different types of cyberbullying behaviours specifically related to different profiles of internet and social media usage?*

In Table 3 are reported the frequencies of the responses to the questions about the time participants typically spend on social media every day, the number of social accounts held, and the number of friends/followers. Approximately half of participants (52.8%) declared to spend every day between 1 and 3 hours using social media. Most participants (69%) reported having only one social account per social media, and only 30% of them had more than one social account. More than half of the respondents (68%) reported having from 200 to 500 followers. For each of these variables, we used Chi-square tests to compare the distributions of responses across gender and age.

Girls reported using social media for more than 3 hours per day (41.5%) significantly ($\chi^2(3)$ = 60.38, p < .001) more often than boys (22.5%). No differences were found between age groups.

Girls also reported having more than one account (33.4%) more frequently ($\chi^2(2)$ = 13.40, p = .001) than boys (23.8%). The distribution of the responses about the number of social network accounts varied significantly across age groups ($\chi^2(6)$ = 45.28, p < .001). Having more than one account tended to be more frequent in the lower age groups (14–15 years: 35.3%; 16–17 years: 30%), than in the other groups (18–20 years: 16.6%, 21–25 years: 15.8%), and in the highest age group having no account at all was significantly more frequent (10.5%) than in the other groups (0.8%-1.4%).

The distribution of the number of followers of each participant varied significantly across gender ($\chi^2(4)$ = 46.52, p < .001): girls had more frequently between 200 and 500 friends/followers (74.6%) than boys (58.9%), while boys had more frequently less than 200 followers (27.7%) than girls (12.6%). No differences were found between age groups.

Whatsapp (91.9%) and instagram (91%) were the most used social networks, equally by boys and girls. YouTube was used by 67% of participants, Facebook by 28.8%, Google+ by

**Table 3. Use of social media.**

| Variable | N | % |
|---|---|---|
| *Time spent on social media* | | |
| Never | 5 | 0.41% |
| Less than one hour per day | 155 | 12.65% |
| From 1 to 3 hours per day | 647 | 52.82% |
| More than 3 hours per day | 418 | 34.12% |
| *Number of social accounts* | | |
| No account | 15 | 1.23% |
| One account | 844 | 69.12% |
| More than one account | 362 | 29.65% |
| *Number of followers* | | |
| < 200 | 221 | 18.42% |
| 200–500 | 821 | 68.42% |
| 501–1000 | 88 | 7.33% |
| 1001–5000 | 64 | 5.33% |
| > 5000 | 6 | 0.50% |

Descriptive statistics (frequencies) about the use of social media in the sample (N = 1228).

**Table 4. Product-moment correlations between personality factors, aggressive behaviours, and moral disengagement.**

| | E | A | C | ES | O | MD |
|---|---|---|---|---|---|---|
| 1. sending nasty or indecent messages | 0.07 | **-0.15** *** | **-0.19** *** | -0.02 | 0.04 | **0.34** *** |
| 2. sending nasty or indecent emails | 0.06 | -0.05 | -0.09 | +0.01 | 0.03 | **0.29** *** |
| 3. sending embarrassing photos of someone via cell phone | 0.06 | -0.09 | **-0.15** *** | -0.02 | **0.10** * | **0.16** *** |
| 4. pretending to be someone else on the Internet | 0.01 | **-0.09** * | **-0.20** *** | -0.02 | 0.06 | **0.19** *** |
| | | (p = .045) | | | | |
| 5. telling someone else's secrets online or through a cell phone without permission | 0.06 | -0.06 | **-0.15** *** | -0.03 | 0.04 | **0.13** *** |
| 6. spreading gossip about someone on the Internet | **0.11** ** | **-0.12** ** | **-0.11** ** | -0.01 | 0.04 | **0.19** *** |
| | (p = .009) | (p = .002) | (p = .006) | | | |

E = Extraversion. A = Agreeableness. C = Conscientiousness. ES = Emotional Stability. O = Openness. MD = Moral Disengagement.

*p < .05.

** p < .01.

*** p < .001.

22.9%, Snapchat by 14.2% and Pinterest by 10.9%. Boys reported using YouTube ($\chi^2(1) =$ 30.38, p < .001) more than girls. Girls reported using Pinterest (15,20%, $\chi^2(1) = 36.18$, p < .001), Snapchat (18.53%, $\chi^2(1) = 29.88$, p < .001) and Tumblr (8.27%, $\chi^2(1) = 14.20$, p < .001) more than boys (Pinterest: 4.20%, Snapchat: 7.35%, Tumblr: 2.94%).

## Correlations between cyberbullying and personality

*RQ3. Are different types of cyberbullying behaviours specifically related to different profiles of internet and social media usage?*

To investigate the relationships between cyberbullying perpetration, personality traits and other individual variables we first computed a correlational analysis. Table 4 shows Pearson product-moment correlations between personality factors, aggressive behaviours, and moral disengagement about cyberbullying.

Among the personality traits, conscientiousness was significantly and negatively correlated with all the types of cyberbullying but sending mean email. Agreeableness was negatively correlated with sending mean messages, pretending to be someone else and spreading gossip, while openness and extraversion were positively correlated with, respectively, sending embarrassing photos and pretending to be someone else.

Moral disengagement was positively and significantly correlated with all the aggressive behaviours, as expected (*r* ranging from .16 to .34).

## Predictors of cyberbullying behaviours

We investigated the predictors of the self-reported frequency of perpetration of the 6 different cyberbullying behaviours using 6 multiple regression models. We considered 12 variables (demographics, personality, use of social networks, moral disengagement, and internet addiction) as possible predictors of cyberbullying, using treatment coding for all the non-numerical variables, and entering simultaneously all the predictors. Since three categorical variables have 3 levels, each of them was coded using two binary predictors, and the total number of predictors in the models was therefore 15. The analyses were conducted after filtering out data from participants that reported to never use social media, or to not have any accounts on them, or that did not provide answers to the items about cyberbullying perpetration. The few

**Table 5. Results of the ordinal regressions models for the responses about the frequency of different cyberbullying behaviours.**

| | 1: Sending mean / indecent messages | | 2: Sending mean / indecent emails | | 3: Sending embarrassing photos | | 4: Pretending to be someone else | | 5: Revealing secrets | | 6: Spreading rumors/gossip | |
|---|---|---|---|---|---|---|---|---|---|---|---|---|
| *Predictors* | *OR* | *p* | *OR* | *p* | *OR* | *p* | *OR* | *p* | *OR* | *p* | *OR* | *p* |
| Gender: M | 2.35 | **<0.001** | 1.47 | 0.235 | 1.32 | **0.045** | 0.80 | 0.199 | 0.88 | 0.390 | 0.93 | 0.652 |
| Age: 16–17 years (vs 14–15 years) | 1.20 | 0.216 | 0.77 | 0.454 | 0.93 | 0.614 | 0.91 | 0.577 | 0.97 | 0.815 | 1.20 | 0.273 |
| Age: 18–20 years (vs 14–15 years) | 1.36 | *0.074* | 1.36 | 0.422 | 1.14 | 0.417 | 0.91 | 0.633 | 0.93 | 0.650 | 1.00 | 0.992 |
| Agreeableness | 0.78 | **<0.001** | 0.85 | 0.264 | 0.90 | *0.098* | 0.92 | 0.287 | 0.95 | 0.428 | 0.82 | **0.009** |
| Extraversion | 1.05 | 0.509 | 1.19 | 0.271 | 1.04 | 0.556 | 0.90 | 0.191 | 1.12 | *0.089* | 1.23 | **0.008** |
| Conscientiousness | 0.79 | **<0.001** | 0.86 | 0.283 | 0.81 | **0.001** | 0.73 | **<0.001** | 0.83 | **0.005** | 0.88 | *0.099* |
| Emotional Stability | 0.95 | 0.490 | 1.01 | 0.931 | 1.04 | 0.555 | 1.14 | 0.103 | 1.09 | 0.204 | 1.05 | 0.568 |
| Openness | 1.10 | 0.141 | 1.08 | 0.598 | 1.24 | **<0.001** | 1.14 | *0.074* | 1.13 | **0.049** | 1.12 | 0.137 |
| N. Followers: < 200 (*vs* 200–500) | 0.83 | 0.291 | 0.98 | 0.951 | 0.61 | **0.004** | 0.67 | *0.059* | 0.59 | **0.004** | 0.75 | 0.184 |
| N. Followers: > 500 (*vs* 200–500) | 1.01 | 0.967 | 0.71 | 0.431 | 0.86 | 0.409 | 0.96 | 0.847 | 0.74 | 0.106 | 1.28 | 0.228 |
| Time on SN: < 1 h/day (*vs* 1–3 h/d) | 1.09 | 0.680 | 0.35 | 0.171 | 0.68 | *0.063* | 0.62 | 0.103 | 0.56 | **0.012** | 0.82 | 0.452 |
| Time on SN: > 3 h/day (*vs* 1–3 h/d) | 1.44 | **0.013** | 1.11 | 0.754 | 1.00 | 1.000 | 1.00 | 0.990 | 1.08 | 0.573 | 1.17 | 0.332 |
| N. of SN accounts: > 1 (*vs* 1) | 2.15 | **<0.001** | 1.62 | 0.113 | 1.60 | **0.001** | 2.98 | **<0.001** | 1.46 | **0.007** | 2.03 | **<0.001** |
| Internet addiction | 1.07 | 0.238 | 1.17 | 0.246 | 1.17 | **0.005** | 1.16 | **0.024** | 1.24 | **<0.001** | 1.06 | 0.358 |
| Moral Disengagement | 1.78 | **<0.001** | 2.07 | **<0.001** | 1.28 | **<0.001** | 1.45 | **<0.001** | 1.24 | **0.001** | 1.48 | **<0.001** |
| Observations | 1120 | | 1120 | | 1120 | | 1120 | | 1120 | | 1120 | |
| $R^2$ Nagelkerke | 0.226 | | 0.163 | | 0.111 | | 0.1777 | | 0.106 | | 0.126 | |

The values in this table correspond to the Odds Ratios (OR, and associated p-values) for the effects of different predictors on the self-reported frequency of 6 different cyberbullying behaviours, estimated by fitting (for each behaviour) an ordinal multiple regression. All the predictors were entered simultaneously in the regression models, in which the dependent variables were ratings of the reported frequency of the perpetration of the behaviours by the respondents, expressed on a 5-point frequency scale (from "never" to "more than 5 times"). SN = Social Networks.

participants in the 21–25 age group were also excluded. Table 5 shows the results of the analysis, in the form of Odds Ratio (OR) and associated p-value for all the predictors, and the estimates of the percentage of variance explained by each model (Nagelkerke's $R^2$). Being multinomial (ordered) models, an OR significantly greater than 1 for a predictor indicates that the probability that the respondent declared to have never perpetrated a specific form of cyberbullying was associated with, and tended to *decrease* across, the levels of the predictor (whether it is continuous or discrete). While the probability of having perpetrated the given behaviour with any given frequency, as opposed to never or a lower frequency, tended to *increase*. The opposite response pattern concerning the frequency of cyberbullying perpetration corresponds to an OR significantly lower than 1. The response probabilities of the different frequencies of perpetrating various forms of cyberbullying, as a function of the significant predictors in the regression analyses, are reported in a series of plots in S1 File.

As it can be seen in the table, among the predictors considered in the analysis only moral disengagement was significantly associated with *all* the forms of cyberbullying (with OR ranging from 1.24 to 1.78, p < .001). Different predictors were in fact associated with different behaviours.

Age was not associated with any behaviour. Gender was associated with *sending mean or indecent messages*, whose odds were 2.35 times higher for boys than for girls (p < .001), and with *sending embarrassing photos* (OR = 1.32, p = .045). Other significant predictors for this behaviour, both with ORs smaller than 1 indicating a protective effect, were conscientiousness (OR = 0.79, p < .001) and agreeableness (OR = 0.78, p < .001). Besides these personality traits

the results of the analysis showed that the likelihood of *sending mean messages* was significantly higher for those spending more than 3 h/d on social networks (OR = 1.44, p = .013) than for those spending 1 to 3 h/d on them, and for those who reported to have more than one profile on social networks (OR = 2.15, p <. 001). Overall, the model explained about 23% of the variance in the frequency of this type of cyberbullying.

The likelihood of *sending mean or indecent emails* was only significantly predicted by moral disengagement (OR = 2.02, p < .001). *Sending embarrassing photos* was less likely perpetrated by girls, individuals with higher conscientiousness (OR = 0.81, p = .001) and fewer followers (OR = 0.61, p = .004), and was more likely perpetrated by those with higher openness (OR = 1.24, p < .001), having multiple accounts on some social networks (OR = 1.60, p = .001) and higher addition to the internet (OR = 1.17 p = .005).

*Pretending to be someone else online* when interacting on social networking sites was less likely in more conscientious individuals (OR = 0.73, p < .001) and more likely in those having multiple accounts on some social networks (OR = 2.98, p < .001) and higher addition to the internet (OR = 1.16 p = .024).

*Revealing secrets about someone online* was also less likely in more conscientious individuals (OR = 0.83, p = .005) and in those having less than 200 followers (OR = 0.59, p = .004) and spending less than 1h/d on social networks (OR = 0.56, p = .012), and more likely in more open individuals (OR = 1.13, p = .049), in those who reported more getting nervous without internet (OR = 1.24, p < .001), and in those having more than one profile on social networks (OR = 1.46, p = .007).

Lastly, the likelihood of *spreading rumors or gossip online* was higher in more extraverted individuals (OR = 1.23, p = .008) and in those having more than one profile on some social network sites (OR = 2.03, p < .001), while it was less likely in more agreeable respondents (OR = 0.82, p = .009).

## Latent variable analyses

*RQ4. Is cyberbullying better conceived and measured as a unitary construct or as a multidimensional construct?*

In the light of the results of the regression analysis, we examined the factorial structure of the cyberbullying scale that we used [51], comprising the items with the frequency of different cyberbullying behaviours. We randomly split the dataset into a training subset (30%, N = 336) and a test subset (70%, N = 784). On the training subset we conducted an Exploratory Factor Analysis, using minimal residual for factor extraction and oblimin rotation. Polychoric correlations were used for the analysis, and parallel analysis was used to determine the number of factors to retain. The model extracted comprised two factors, each loading on 3 items (Table 6): one -Cyb1- loaded on *sending mean messages* (loading = 0.66), *sending mean emails* (loading = 0.77), and *impersonating someone else online* (loading = 0.43), and the other one -Cyb2- loaded the remaining three cyberbullying behaviours (*sending embarrassing photos*: loading = 0.42, *revealing secrets*: loading = 0.93, *spreading rumors/gossip*: loading = 0.60). The factors were positively correlated (r = 0.43), and together explained 52% of the variance.

On the test set we then conducted two Confirmatory Factor Analyses (CFA), one using the single factor model used in the Meter and Bauman scale [51], and the other using the 2-factor model suggested by the EFA. Both models had a very good fit to the data, but the 2-factor model had slightly better fit according to all the fit measures used [1 factor: $Chi^2$(9) = 19.027, p = .025, N = 784, CFI = 0.991, NFI = 0.984, NNFI = 0.986; 2-factor: $Chi^2$(8) = 11.175, p = 0.192, CFI = 0.997, NFI = 0.991, NNFI = 0.995].

**Table 6. Results of the exploratory factor analysis of cyberbullying perpetration.**

| Cyberbullying behaviour (item) | Cyb1 | Cyb2 | u |
|---|---|---|---|
| 1. Sending nasty or indecent messages | 0.66 | | 0.55 |
| 2. Sending nasty or indecent emails | 0.77 | | 0.44 |
| 3. Sending embarrassing photos of someone via cell phone | | 0.42 | 0.58 |
| 4. Pretending to be someone else on the Internet | 0.43 | | 0.65 |
| 5. Telling someone else's secrets online / by cell phone without permission | | 0.93 | 0.19 |
| 6. Spreading gossip about someone on the Internet | | 0.60 | 0.49 |
| Proportion of variance explained (R²) | 0.25 | 0.27 | |

Results of the exploratory factor analysis of the cyberbullying perpetration items behaviour items, conducted on a randomly selected subset of participants (N = 336). For each item we report the loadings on each of the two latent factors extracted in the analysis and the proportion of unique item variance ($u$). For each factor we also report the relative proportion of explained variance. The loadings presented correspond to the rotated solution (pattern matrix). Factor loadings < .4 were omitted from the table.

The latent cyberbullying factors were used in a Structural Equation Modelling analysis (SEM) to test hypotheses about the direct and indirect effects of personality, moral disengagement, and internet addiction on the different types of cyberbullying. The structural model tested in the analysis is presented in Fig 3 along with the standardised estimates of all the path coefficients (β) (model fit statistics are reported in the caption).

As it can be seen, and consistently with the previous analyses, each of the cyberbullying factors was associated with some, but not all, of the BFI personality traits. And whenever a trait was significantly associated with both cyberbullying factors, the strength of association (as measured by the standardised path coefficients) varied across factors. In most cases, the strength of the association was greater for the first factor than for the second one. This was particularly the case for the positive effect of moral disengagement, but also for the negative (i.e. protective) effect of conscientiousness and agreeableness. Conversely, the personality factors that were *positively* associated with cyberbullying tended to be associated only (i.e. extraversion) or more strongly (i.e. openness) with the second cyberbullying factor. Overall, the model was able to explain 50% of the variance in the first latent cyberbullying factor, and 24% in the second one.

Given that all the personality traits but agreeableness were also significantly associated with moral disengagement, explaining about 4% of the variability in this latent variable, we tested the indirect effects of each personality factor on the cyberbullying factors mediated by their effect on moral disengagement. In Table 7 are reported the results of these tests. As it can be seen in the table, all the personality factors that were directly associated with moral disengagement had a significant indirect effect on the cyberbullying factors. The tests of the total effects of the personality factors on cyberbullying were significant except for the one of emotional stability, and confirmed that agreeableness and conscientiousness tended to reduce cyberbullying, particularly of the type involving direct aggression toward others online (cyberbullying factor 1), while openness and extraversion tended to increase cyberbullying, particularly of the type involving damages to someone's reputation or public image (cyberbullying factor 2).

To test differences between groups defined by gender (females vs males), time on social networks (+3 h/d vs 1–3 h/d) and number of profiles on social networks (1 profile vs >1 profile) in the mean score of the latent factors (the two cyberbullying factors and moral disengagement), and to test possible moderating effects of each of the grouping factors on the direct and indirect effects of personality and moral disengagement on the cyberbullying factors, three

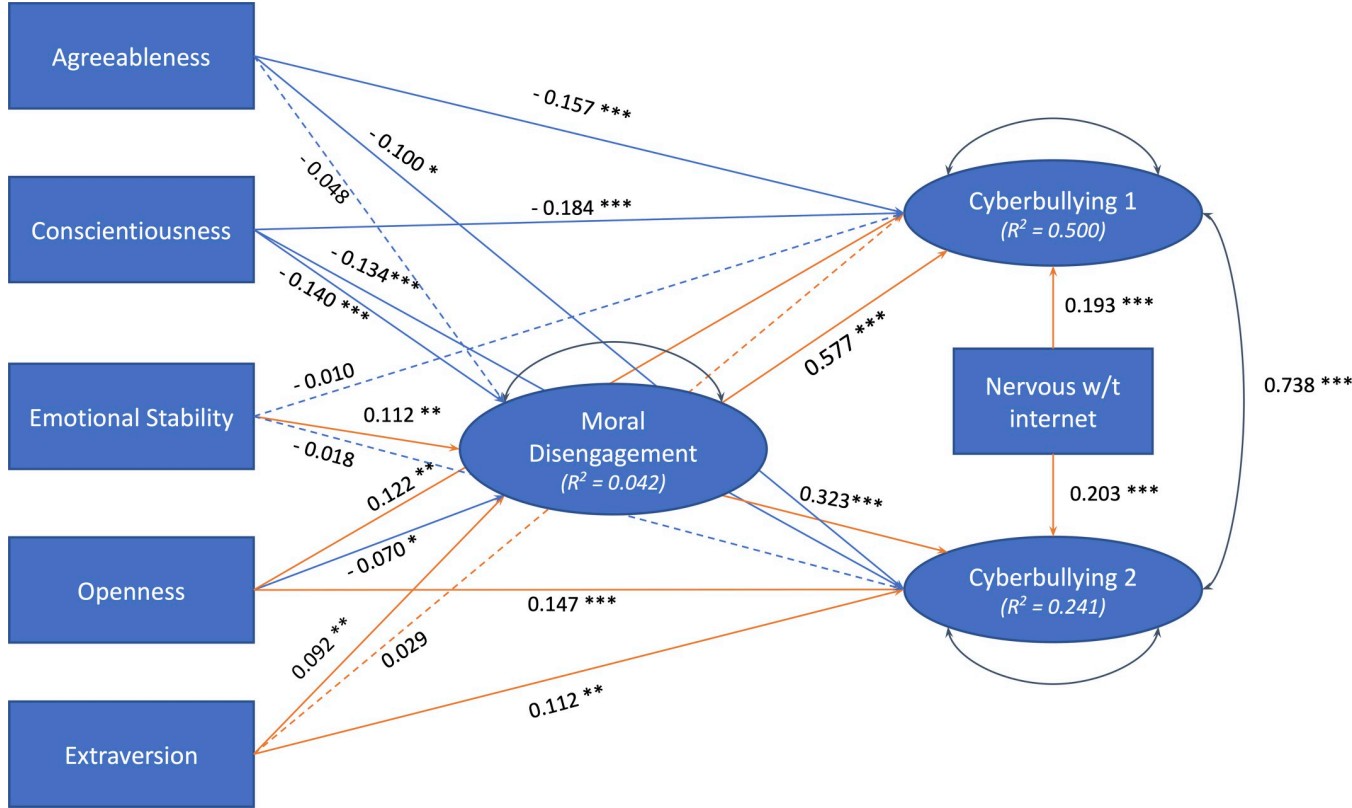

**Fig 3. Structural equation model of the relationships between personality, moral disengagement, and cyberbullying factors.** Ovals and rectangles represent latent and manifest variables respectively. Standardized path coefficients are reported for all the direct effects tested in the model, and the colours of the paths represent the sign of the coefficients (orange = positive; blue = negative). Insignificant effects are represented by dashed arrows. [N = 1139, $\chi^2(141)$ = 358.64 (p = .000); RMSEA = .037 (90% CI = [.032, .042]); NNFI = .977; CFI = .964].

multigroup SEM analyses were conducted, one for each of the grouping variables. In these analyses, first we always tested measurement invariance (configural, weak and strong), since comparisons across the groups in the mean scores of the latent variables and in the regression paths could only be meaningful if strong invariance holds.

The results of invariance tests, and the parameters of the multigroup models for each group and grouping factor are provided in S2 File. Strong measurement invariance held across groups for each grouping factor, indicating that it was possible to compare latent variables means and effects estimates across groups.

The multigroup analysis about gender revealed significantly higher mean levels of moral disengagement (diff. = 1.782, p = .010) for males than for females. Moreover, the results showed that gender significantly moderated both the effect of openness on the first cyberbullying factor (slope $_{females}$ = 0.198, slope $_{males}$ = 0.029, diff. = -0.169, p = .026), and the effect of extraversion on the second cyberbullying factor (slope $_{females}$ = 0.047, slope $_{males}$ = 0.219, diff. = 0.171, p = .034).

The mean of moral disengagement was significantly higher in those spending 1–3 hour per day on SN than in those spending more than 3 h/d (diff. = 1.168, p = .039), and time on SN was found to be a moderator of the direct effect of emotional stability on moral disengagement (slope $_{+3 h/d}$ = 0.217, slope $_{1-3 h/d}$ = 0.005, diff. = 0.212, p = .005) and of the indirect effects of this personality trait on the cyberbullying factors (cyberbullying factor. 1: diff. = 0.133, p = .004; cyberbullying factor 2: diff. = 0.071. p = .025).

**Table 7. Results of the mediation analysis from the structural equation modelling analysis.**

| Direct effects | | | Indirect effects | | | Total effects | | |
|---|---|---|---|---|---|---|---|---|
| Effect | Est. | p | Effect | Est. | p. | Effect | Est | p. |
| A → CBY 1 | **-0.157** | < .000 | A → MD → CYB 1 | -0.030 | 0.139 | A → CBY 1 | **-0.186** | < .000 |
| C → CYB 1 | **-0.184** | < .000 | C → MD → CYB 1 | **-0.076** | < .000 | C → CYB 1 | **-0.260** | < .000 |
| ES → CBY 1 | -0.010 | 0.795 | ES → MD → CYB 1 | **0.065** | 0.002 | ES → CBY 1 | 0.054 | 0.191 |
| O → CYB 1 | **0.122** | 0.002 | O → MD → CYB 1 | **-0.040** | 0.033 | O → CYB 1 | **0.081** | 0.048 |
| E → CYB 1 | 0.029 | 0.454 | E → MD → CYB 1 | **0.053** | 0.009 | E → CYB 1 | **0.082** | 0.041 |
| A → CBY 2 | **-0.100** | 0.011 | A → MD → CYB 2 | -0.017 | 0.146 | A → CBY 2 | **-0.117** | 0.003 |
| C → CYB 2 | **-0.140** | < .000 | C → MD → CYB 2 | **-0.042** | 0.001 | C → CYB 2 | **-0.182** | < .000 |
| ES → CBY 2 | –0.018 | 0.650 | ES → MD → CYB 2 | **0.036** | 0.004 | ES → CBY 2 | 0.018 | 0.633 |
| O → CYB 2 | **0.147** | < .000 | O → MD → CYB 2 | **-0.023** | 0.040 | O → CYB 2 | **0.124** | 0.001 |
| E → CYB 2 | **0.112** | 0.005 | E → MD → CYB 2 | **0.030** | 0.015 | E → CYB 2 | **0.141** | < .000 |
| A → MD | -0.048 | 0.135 | | | | | | |
| C → MD | **-0.134** | < .000 | | | | | | |
| ES → MD | **0.112** | 0.001 | | | | | | |
| O → MD | **-0.070** | 0.033 | | | | | | |
| E → MD | **0.092** | 0.008 | | | | | | |
| MD → CYB 1 | **0.577** | < .000 | | | | | | |
| MD → CYB 2 | **0.323** | < .000 | | | | | | |

Lastly, the results of the multigroup analysis involving the number of social network profiles revealed significantly higher mean levels of moral disengagement in those with more than one profiles than in those with only one social network profile (diff. = 1.270, p = 0.045), but no differences in the mean of the latent cyberbullying factors. Moreover, the number of profiles significantly moderates the direct effect of extraversion on moral disengagement (slope $_{1\ profile}$ = 0.142, slope $_{>1\ profiles}$ = -0.022, diff. = 1.64, p = .020) and the indirect effect of this personality factor on the cyberbullying factors mediated by moral disengagement (cyberbullying factor. 1: diff. = 0.98, p = .040; cyberbullying factor 2: diff. = 0.059. p = .043).

## Discussion

The results of our study showed that different types of aggressive online behaviours are, reportedly, enacted with differential frequency by adolescents, confirming H1. For all the forms of cyberbullying considered, most participants reported they have never committed them, and for some behaviours, such as sending mean or aggressive emails, even reports of having perpetrated them sometimes were extremely rare. This last result is possibly related to uncommon use of emails among adolescents [74]. But other types of cyberbullying, like aggressions by text messages or sending embarrassing photos of someone, were perpetrated at least sometimes by more than 40% of participants, and often (≥ 3 times) by 13–18%. Gender differences in the distribution of cyberbullying frequency were found only for some specific forms of cyberbullying, again sending mean messages or embarrassing photos, therefore H2 is also confirmed by the results. However, differently from other studies (see, for example, [54, 75, 76]) no differences across age groups were found. This result is probably due to the narrow age range of our sample that did not include children, in which young adults were almost entirely absent.

Moral disengagement on cyberbullying was strongly associated with all forms of cyberbullying, confirming previous findings [13, 51, 52], while personality was not: *conscientiousness* was the personality trait that was associated with the highest number of cyberbullying behaviours (4 out of 6, all but *sending mean emails* and *spreading rumor/gossip*), in each case having a

protective effect (i.e. more conscientious individuals reported less frequent perpetration) [77]. Agreeableness also had a protective effect [42], but only on sending mean messages. Openness was instead positively associated with sending mean messages, sending embarrassing photos and revealing secrets, and extraversion with spreading rumors/gossip. Emotional stability was not associated with any behaviours, in contrast to what has been previously reported by one study [65]. However, in the SEM analysis we found significant positive indirect effects of emotional stability on both cyberbullying factors, mediated by moral disengagement. The findings about a positive association between emotional stability and moral disengagement also contrast with previous findings [78]. The SEM analysis also confirmed that the effect of moral disengagement was stronger on cyberbullying acts involving direct aggressions than on acts damaging social image or reputation. Regarding the relationships between personality traits and different cyberbullying behaviours, it is interesting to note that no two behaviours are associated with the same traits. This suggests that the personality structures underlying the different behaviours may be quite specific (H3).

The results regarding the association between time spent online and engaging in virtual offensive behaviours is particularly relevant as it becomes clearer and clearer how internet use is increasing across cultures [79]. The ordinal regression analysis showed, first of all, that the daily amount of time spent on social networks is predictive only of some cyberbullying behaviours, not all, confirming H4. This specific relationship between the frequency of internet use and specific cyberbullying behaviours had not been highlighted by other studies that, instead, had indicated a generic relationship [42, 68]. Specifically, in our findings the likelihood of sending mean messages (with some frequency) tended to be higher in those that spend more than 3 h/d on social networks than in those that spend between 1 and 3 h/d, while the likelihood of sending embarrassing photo and revealing secrets tented to be lower in those that use social networks less than 1 h/d. Moreover, the results of the multigroup SEM analysis showed that time on social networks moderated the indirect effect of emotional stability on both the cyberbullying factors, which was only significant and positive (opposite in sign to what previous research would have let to expect) for those spending more than 3 h/d on social networks.

Lower likelihood of perpetrating the latter two behaviours was also found in individuals with few followers/friends on social networks. It is possible that these factors were associated with cyberbullying because spending more or less time online, or having a wider or a smaller social network simply increases or decreases the *occasions* for acts of cyberbullying. However, it is not clear why this association was only found with these specific behaviours and not for all. Among the social media use variables that we considered, at least one was however positively and strongly associated with almost all the cyberbullying behaviours: having more than 1 profile on social networks, a factor that increased the odds of cyberbullying between 46% and 198%, depending on the behaviour considered. Only the likelihood of sending mean/indecent emails was not associated with this factor. It might be that having more profiles on social networks is an enabling factor for cyberbullying, a means for perpetrating aggressive acts online in disguise trying to escape possible consequences of the acts. It is worth noting that the strongest association was found between this factor and pretending to be someone else online. But it is also possible that some other individual factor, not measured in the current study, might be responsible for the association. It is also interesting to notice that the multigroup SEM analysis showed higher levels of moral disengagement in those having more than 1 profile on social networks, and also that this factor moderated (increased) the indirect effects of extraversion on cyberbullying mediated by moral disengagement. Further studies should try to investigate this matter more in depth.

Lastly, our analysis highlighted an association between internet addiction and some bullying behaviours [77], such as sending embarrassing photos, revealing secrets, and pretending to

be someone else online, whose frequency was higher in individuals that tended to be nervous without access to the internet. These are three behaviours that, in addition to moral disengagement, are all associated with a low level of conscientiousness. This makes it possible to consider a relationship between the diminished capacity for self-control, self-regulation and impulse control and forms of internet addiction. This consideration is in line with what has been highlighted by a meta-analysis in which it emerges that among the factors that influence internet addiction, in addition to the measuring instrument used, and the cultural area of reference, there is the young age of the participants [80]. Unfortunately, however, in our study, the degree of internet addiction was measured with a single item. Therefore, it is incautious to go into more fine-grained explanations of the relationship between internet addiction and the enactment of specific aggressive virtual behaviours.

Our last research question concerned the measurement of cyberbullying, and specifically whether it should be conceived and measured as a unitary or a multidimensional (H5) construct. The results of our latent variable analyses all seem to point toward a multivariate nature of cyberbullying, confirming H5. The exploratory factor analysis suggests that one could distinguish between a component related to harmful acts that an individual might commit to someone else directly (e.g. offending the target by direct messages, emails, or deceiving someone about their identity), and a component related to acts that might hurt someone even without a direct interaction and by damaging the target's social image (i.e. acts in which harm require the involvement of a social audience). And the confirmatory factor analysis showed that the 2-factor structure has better fit than the single factor solution. Interestingly, the results of the SEM analysis suggested that these factors might be differentially affected by personality and moral disengagement. Direct aggressive acts (Cyberbullying factor 1) are more strongly associated with moral disengagement, with having multiple SN profiles, and with personality traits such as conscientiousness and agreeableness (both playing a mitigating role) than acts that harm someone indirectly (Cyberbullying factor 2). The latter form of cyberbullying, conversely, is particularly associated with extraversion and openness. Moreover, only for direct aggressive acts we found differences between female and male cyber-aggressors. Overall, these findings suggest that indeed cyberbullying might not be a unitary phenomenon, as other studies also had indicated [34, 43, 62, 63, 81–83] and that differences in the findings from the research literature about the association between cyberbullying and various demographic and individual variables could be the results of differences in the specific measurement of cyberbullying used across studies. Differentiating distinct forms of cyberbullying with different antecedents or moderators could help to identify in early evaluations which subjective and contextual characteristics can more easily lead to the emergence of specific offensive virtual acts. And from this, can descend not only the understanding of the more detailed forms of cyberbullying, but also the design of interventions aimed more effectively and more promptly at its prevention and mitigation.

## Limitations

The study reported in this paper has some limitations that need to be acknowledged. The first limitation concerns the type of measurement of cyberbullying used in the study, which was self-reported, and therefore possibly affected by biases such as the social desirability effect, the tendency to provide responses that give a better impression of oneself as a person and member of society [84, 85]. When it comes to reporting the frequency of perpetrating aggressive behaviours, it is likely that this phenomenon might have been at play, and, in this study, it was not controlled for. Even if that was the case, however, this bias should not undermine our findings about predictors of cyberbullying, but only cause our estimates of the frequency of the

phenomenon among adolescents to be lower than the actual frequency. Nonetheless, repeating this study using observational data as an outcome, possibly collected from adolescents' activity on social networks (posts, tweets, comments, photos or other forms of shares directly or indirectly involving a victim), would increase the validity of the results and the confidence in our findings.

The second limitation of this study lies in the cross-sectional nature of the survey, which only allows us to draw conclusions about statistical associations between individual variables and cyberbullying, preventing us from being able to infer causal relationships. While for ethical reasons studying this phenomenon experimentally would be very difficult, if not unfeasible, at least a longitudinal design could partly help to overcome this limitation concerning internal validity.

A third limitation is related to the way some variables were measured. On the one hand, in fact, one of the predictors that were statistically associated with more bullying behaviours was internet addiction, which was only measured with a single item on a 5-point Likert scale. The effect of this variable on cyberbullying should be thus investigated more in depth, adopting more articulated measures, to validate and possibly extend our findings. On the other hand, even the scale we adopted to measure cyberbullying was quite specific and considered a limited number of behaviours only. This is an important limitation of the study. It is still to be determined, and certainly worth investigating, whether our findings about the multidimensional nature of cyberbullying would be found also using different measures and scales.

Lastly, our study was conducted on a sample of Italian adolescents, and although there is no reason to believe that our findings shouldn't generalize to different cultural contexts, further studies should still be conducted on an international sample to empirically test this hypothesis.

## Conclusions

Considering recent knowledge advances on cyberbullying, including the findings from the study presented in this paper, it seems reductive to qualify this offensive behaviour as a unitary phenomenon. Rather, considering the different variables involved, it seems more likely that there are different expressions of cyberbullying, offensive behaviours that can be differentiated by considering socio-demographic variables, personality characteristics, and the different relational tools available on the Internet.

The study presented here considered a scale with six items. For each of these items, which correspond to specific offending behaviours, different correlates of age, personality, and social media interaction were highlighted. From our results, it was also possible to put forward a conceptual framework that describes cyberbullying as a harmful behaviour that can have at least two facets: one related to offenses perpetrated directly to the victim, the other less explicit, which can involve other people and is indirect. It is legitimate to consider that with different scales, and with a greater number of items, it would be possible to point out further aspects of cyberbullying. In any case, the results obtained and the theoretical model proposed call for deeper investigations of cyberbullying, not only for its different manifestations, but also for individual and contextual reasons that make it occurring.

## Supporting information

**S1 Text. Questionnaire.** Full text of questionnaire used in the study.
(PDF)

**S1 Script. R script for conducting the descriptive statistics reported in the paper.**
(R)

**S2 Script. R for conducting the regression analyses reported in the paper.**
(R)

**S3 Script. R script for conducting the exploratory and confirmatory factor analyses reported in the paper and the latent variable path analysis / SEM.**
(R)

**S1 Dataset. R Dataset for descriptive statistics.** Data are in native R format.
(RDATA)

**S2 Dataset. R Dataset used for the regression and SEM analyses.** Data are in native R format.
(RDATA)

**S3 Dataset. Dataset used for the regression and SEM analyses.** Data are in native comma separated value format.
(CSV)

**S1 File. Plots of the effects of the significant predictors in the ordinal logistic regression models.** Each plot represents the estimated probability of the different responses concerning the frequency of perpetration of a given type of cyberbullying behaviour, as a function of one of the significant predictors. The probabilities were estimated from the fitted regression models presented in the paper. All the code for reproducing the plots is provided in S2 File.
(PDF)

**S2 File. Multigroup SEM analyses.** The results of the multigroup analyses for three grouping factors are reported in this file: a) gender (females vs males), b) time on social network (>3 h/d vs 1–3 h/d), and number of social networks profiles (1 profile vs >1 profiles). For each factor we present the goodness-of-fit statistics of models assuming different levels of invariance and the path coefficients for the model assuming strong invariance.
(PDF)

## Acknowledgments

We would like to thank all those who contributed to the realization of this work: the high school students, the teachers, and the school principals. A special thanks is also due to the students of the Cognitive Psychology course for their helpful contribution in collecting the data, and to Antonio Rizzo for his concrete support in order to see this work published.

## Author Contributions

**Conceptualization:** Stefano Guidi, Paola Palmitesta, Oronzo Parlangeli.

**Data curation:** Stefano Guidi, Ileana Di Pomponio.

**Formal analysis:** Stefano Guidi, Ileana Di Pomponio.

**Funding acquisition:** Paola Palmitesta, Oronzo Parlangeli.

**Investigation:** Stefano Guidi, Paola Palmitesta, Margherita Bracci, Enrica Marchigiani, Oronzo Parlangeli.

**Methodology:** Stefano Guidi, Paola Palmitesta, Margherita Bracci, Enrica Marchigiani, Oronzo Parlangeli.

**Project administration:** Paola Palmitesta, Margherita Bracci, Enrica Marchigiani, Oronzo Parlangeli.

**Supervision:** Paola Palmitesta, Oronzo Parlangeli.

**Validation:** Stefano Guidi.

**Visualization:** Stefano Guidi.

**Writing – original draft:** Stefano Guidi, Margherita Bracci, Enrica Marchigiani, Ileana Di Pomponio, Oronzo Parlangeli.

**Writing – review & editing:** Stefano Guidi, Paola Palmitesta, Margherita Bracci, Enrica Marchigiani, Oronzo Parlangeli.

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
