## [Decision Letter · Decision Letter 0]

4 Feb 2022

PONE-D-21-37456

How many cyberbullying(s)? A non-unitary perspective for offensive online behaviours

PLOS ONE

Dear Dr. Guidi,

Thank you for submitting your manuscript to PLOS ONE. After careful consideration, we have decided that your manuscript does not meet our criteria for publication and must therefore be rejected.

Specifically, the reviewers expressed concerns with the methodology, data analysis and reports of the findings. 

I am sorry that we cannot be more positive on this occasion, but hope that you appreciate the reasons for this decision.

Yours sincerely,

Mingming Zhou, Ph.D.

Academic Editor

PLOS ONE

Reviewers' comments:

Reviewer's Responses to Questions

**Comments to the Author**

1. Is the manuscript technically sound, and do the data support the conclusions?

Reviewer #1: Partly

Reviewer #2: Partly

Reviewer #3: Yes

2. Has the statistical analysis been performed appropriately and rigorously? 

Reviewer #1: Yes

Reviewer #2: No

Reviewer #3: Yes

3. Have the authors made all data underlying the findings in their manuscript fully available?

Reviewer #1: Yes

Reviewer #2: Yes

Reviewer #3: Yes

4. Is the manuscript presented in an intelligible fashion and written in standard English?

Reviewer #1: Yes

Reviewer #2: No

Reviewer #3: Yes

5. Review Comments to the Author

Reviewer #1: Dear author and editor,

I consider this research to be rigorous and useful to the scientific community.

However, we should improve some aspects to favor the reader's understanding. These are minor revisions.

Abstract: add all sociodemographic data, sex and age.

Theoretical framework: add more meta-analysis references that support your theoretical framework. Here are some studies that could be interesting:

Pan, Y. C., Chiu, Y. C., & Lin, Y. H. (2020). Systematic review and meta-analysis of epidemiology of internet addiction. Neuroscience & Biobehavioral Reviews, 118, 612-622. https://doi.org/10.1016/j.neubiorev.2020.08.013.

Modecki, K. L., Minchin, J., Harbaugh, A. G., Guerra, N. G., & Runions, K. C. (2014). Bullying Prevalence Across Contexts: A Meta-analysis Measuring Cyber and Traditional Bullying. Journal of Adolescent Health, 55(5), 602-611. https://doi.org/10.1016/j.jadohealth.2014.06.007

Marciano, L., Schlz, P. J., & Camerini, A. L. (2020). Cyberbullying perpetration and victimization in youth: A meta-analysis of longitudinal studies. Journal of Computer-Mediated Communication, 25(2), 163-181. https://bit.ly/3iK8b0g

Lei, H., Li, S., Chiu, M. M., & Lu, M. (2018). Social support and Internet addiction among mainland Chinese teenagers and young adults: A meta-analysis. Computers in Human Behavior, 85, 200-209. https://doi.org/10.1016/j.chb.2018.03.041. https://doi.org/10.1016/j.chb.2018.03.041

Lozano-Blasco, R., Cort'es-Pascual, A., & Latorre-Martínez, M. P. (2020). Being a cybervictim and a cyberbully - The duality of cyberbullying: A meta-analysis. Computers in Human Behavior, 111, 106444. https://doi.org/10.1016/j. chb.2020.106444.

1. Lozano-Blasco, R.; Cortés-Pascual, A. Problematic Internet uses and depression in adolescents: A meta-Analysis. Comunicar 2020, 28, 109-120. [Google Scholar] [CrossRef].

Holt, M. K., Vivolo-Kantor, A. M., Polanin, J. R., Holland, K. M., DeGue, S., Matjasko, J. L., Wolfe, M., & Reid, G. (2015). Bullying and Suicidal Ideation and Behaviors: A Meta-Analysis. PEDIATRICS, 135(2), e496-e509. https://doi.org/10.1542/peds.2014-1864.

Methodology: add a main figure where the procedure and research design are visually explained.

Participants: write the sociodemographic information in a table to make it more visual. Clarify the regions that have been involved in Italy, a heat map would bring innovation and creativity.

Instruments: very well developed but do not forget to put the Italian version next to the name. In the references it is fine, but it is important for the scientific community to see that you have used standardized tests adapted to their culture.

Statistical analysis: very well developed, do you think that mediation analysis could provide more information on the moderating variables?

Results: very interesting but Figure 1 has poor visual quality. The concepts are very good but it looks very blurry and the size of the numbers is very small. The same happens with figure 2. Try to use some application like Canva, Flourish that allows you to improve the quality.

Discussion: I liked it very much, but to give it a plus of quality I recommend that you review more references on the use of social networks and cyberbullying. Add more meta-analysis studies that allow you to give more robustness to your claims. Adding the hypotheses in the text was a great idea. The limitations are very sincere and clear, but I miss two paragraphs on practical implications this research would have.

Conclusions: they are great, but you usually don't put references in this section, could you upload those references to discussion and rewrite those sentences in a more general way?

Supplementary material: it is very honest that you upload the questionnaire, but it is an opportunity for this to be used by others. I would encourage you to upload the data of your rating along with it, so that it can be used.

Once again, thank you for your thoroughness and good work.

Reviewer #2: Despite the great efforts, authors are advised to make substantial changes to the manuscript as stated below. Because the review found some limitations regarding methods and results, comments on the discussion section are not addressed.

2. Methods

- Please mention the data collection period and explain the sampling method used

- Were the participants in the study all high school students? It seems like adolescents and young adults are mixed as authors included the age range from 14 to 25.

- Where is the reference to the use of social networks and feelings related to network use questionnaires? Please provide the original reference and how they were modified.

- Are the scales used in the survey all well-validated and -translated in Italian? If not validated, how was it applied to the survey? Did the researcher manually translate the items? If the scales are modified from the original version, please indicate both the original and revised version.

3. Results

- It seems like the results of chi-square is only reported in the text, did the authors not create table?

- Authors attempted to test several hypotheses but having so many hypotheses often mislead the readers to lose focus of the study. To make the results section more reader-friendly, it is recommended for the authors to match the results with the corresponding hypotheses so that they/we could understand why these results are presented in this section.

- When reviewing the methods section, I thought six items used to measure cyberbullying would be combined as a single variable instead of splitting each item. If the purpose of the current study was to measure the different types of cyberbullying, it was more appropriate to choose a scale consisting of several subscales rather than splitting a single measurement tool by items. This is a big limitation therefore it should be clearly stated in the limitation section.

- I don’t quite understand the intention behind conducting the EFA, CFA, and structural equation modelling in this study. This study did not develop a scale nor did it validate the existing scale. Why should these analyses be performed? For what purpose?

- Figure 2 is very confusing, and it is not a typical arrangement for SEM. Please refer back to other SEM research to redraw the figure. Why is there no latent variable for openness, extraversion etc. and for social network uses? Just by looking at this figure, the DV and IV are unclear.

Lastly, the sentences are way too long in some paragraphs. Please polish the language in order to improve the overall quality of the manuscript.

Reviewer #3: This paper conducted a questionnaire that was administered to a large sample of high school students to validate the four hypotheses regarding cyberbullying. These research questions are interesting and provide insights about cyberbullying behavior. For example, statistical analyses in this work show that cyberbullying is not a unitary construct but a multidimensional construct.

I have following suggestions to further improve the manuscript:

1. Given the extreme popularity of AI and Machine Learning, in related work, the authors might consider including recent works that used machine learning models to validate similar findings. For example, in [1], the authors showed that user's personality traits and peer influence are important predictors of cyberbullying. In [2], the authors also showed capturing the repetitive pattern in cyberbullying behavior can improve the performance.

2. The figures are very low-quality and it is hard to read. The authors need to replace them with high-resolution figures.

3. The cyberbullying definition is not clear, and sometimes confuses with other similar concept, such as cyber-aggression [3]. The authors need to clarify these differences.

[1] PI-bully: Personalized cyberbullying detection with peer influence

[2]Hierarchical attention networks for cyberbullying detection on the instagram social network

[3]Cyberbullying: Bullying in the digital age

6. PLOS authors have the option to publish the peer review history of their article (what does this mean?). If published, this will include your full peer review and any attached files.

Reviewer #1: **Yes: **Raquel Lozano Blasco

Reviewer #2: No

Reviewer #3: No

- - - - -

---

## [Author Response · Author response to Decision Letter 0]

29 Mar 2022

Response to Academic Editor

C.1. Dear Dr. Guidi,

Thank you for submitting your manuscript to PLOS ONE. After careful consideration, we have decided that your manuscript does not meet our criteria for publication and must therefore be rejected.

Specifically, the reviewers expressed concerns with the methodology, data analysis and reports of the findings. 

R.E.1. We are sincerely puzzled by this editorial decision, which goes against the suggestions of the reviewers, none of which suggested a rejection decision on our manuscript. Actually, two reviewers (Reviewer 1 and 3) only asked for minor revisions (one going as far as describing our research "rigorous and useful to the scientific community", calling the statistical analysis section "very well developed", the results "very interesting" and the conclusions "great"!), and provided each a few suggestions for improving the manuscript, that we have addressed in the revised manuscript attached (in track changes).

The third one (Reviewer 2) requested substantial changes to the manuscript and raised a number of points concerning the methods and results sections, to which we also have responded pointly in the dedicated section and in the revision. 

Additional comments after authors’ query:

[...] C.2 I share the same concern with Reviewer 2 about the methodology and result section. Without validating the selected scales in the first place, it is really hard to say how convincing the findings would be. 

R.E.2. We thank the Editor for clarifying the reasons for her decision in response to our query. We perfectly understand the importance of validation of a measure, and agree with the Editor about it being a requirement. But we do not believe that in our study lack of validity could be an issue. First of all in our case concerns could only apply to the validity of the Italian translation of some scales (the cyberbullying and moral disengagement for cyberbullying), because the original versions had been validated. 

But our own data from this study (CFA), as well as from a previously published study (https://link.springer.com/chapter/10.1007/978-3-030-49570-1_20) by our group (in which SEM models were used showing very good fit of the measurement model for both the bullying and for moral disengagement scales used in the manuscript we submitted to PlosONE) in our view already provide evidence of the validity of the Italian translation we had previously devised and adopted in this study. 

But this wouldn't be the first time that PlosONE published papers concerning studies in which some scales that were not validated in Italian were used. Just to cite a couple of papers I could list this

https://journals.plos.org/plosone/article?id=10.1371/journal.pone.0243194 that used the Italian version of the WHO-5 well-being scale that, while widely used, to our knowledge has never been validated in a dedicated study. 

Or this other study https://journals.plos.org/plosone/article?id=10.1371/journal.pone.0142715 in which the adapted Italian version of a bullying/victimization scale was used, a scale that although used in other studies was never validated (we traced the references two levels down to verify). 

Lastly, we note that Reviewer 2 did not consider that our manuscript should be rejected for lack of validity, but asked for more information, and as we also responded to the reviewer in the dedicated section, we have now revised the manuscript to include more information about this. 

C.3. Also, the examination of the factorial structure of the cyberbully scale (RQ4, which looked to me less as an independent research question but more as a necessary step of the data analysis) found both one-factor and two-factor model made good fits to the model. The adoption of the two-factor model thus needs a stronger justification. Is there any conceptual meaning of each of these two factors? I wonder if the same two-factor structure could be replicated with another sample. Theoretical underpinning would be necessary to support this two-factor model.

R.E.3. Concerning the 2-factor model, we do believe both that the factors have conceptual meaning (as clearly stated in the manuscript, one is mainly related to Direct aggressions toward someone else online, and the other to Behaviours that damage the social image or reputation of someone else) and also that the procedure to test this 2 factor solution is valid and could be replicated in another sample providing the same results. After all, the EFA was conducted on a random subset of a dataset that comprised responses from a wide and geographically diverse sample of Italian students (randomly split in a smaller train and a larger test subsamples). And CFA confirmed the extracted factor structure in another sample. This analytic strategy is quite common in psychology research [1, 2]. It would have been different (and invalid) if we had run the EFA and the CFA on the same sample, but we did not, and our procedure is also indicated as a viable solution in [3]. 

To further test the replicability of the factor structure, we have now repeated 100 times the parallel analysis (to determine the number of factors) and the EFA on 100 random samples of N = 336, extracting in each case 2 factors, and comparing the factors structures extracted (the pattern of loadings) to the one in the model reported in the manuscript. In the majority of cases (55%) parallel analysis suggested 2 factors (1 factor was suggested only in 2% of the cases) and in 81.8% of the 2-factor models one or both factors overlapped with the 2 factors in the model that we reported. In other words, it seems to us that the model that we reported is both conceptually and empirically grounded. Moreover, a 2-factor model like ours could help explain inconsistencies in the findings in literature about the association between cyberbullying and other measures of social media use and personality. 

[1] Igarashi, H., Kikuchi, H., Kano, R. et al. The Inventory of Personality Organisation: its psychometric properties among student and clinical populations in Japan. Ann Gen Psychiatry 8, 9 (2009). https://doi.org/10.1186/1744-859X-8-9

[2] Matsudaira T., Fukuhara, T., Kitamura, T., (2008). Factor structure of the Japanese Interpersonal Competence Scale. Psychiatry and Clinical Neurosciences. 62(2), 142-151. https://doi.org/10.1111/j.1440-1819.2008.01747.x

[3] Fokkema, M., & Greiff, S. (2017). How performing PCA and CFA on the same data equals trouble: Overfitting in the assessment of internal structure and some editorial thoughts on it [Editorial]. European Journal of Psychological Assessment, 33(6), 399–402. https://doi.org/10.1027/1015-5759/a000460

C.4. It also appears that this two-factor model was not well reflected in Table 3 and 4.

R.E.4. We understand that in Table 4 and 5 (formerly Table 3 and 4) a clear pattern in cyberbullying association with demographic, personality and social media use variables is difficult to highlight. However, in Table 4 items 1 and 4 (both in Factor 1) show the same pattern of correlations with personality and MD, and in Table 5 item 3 and 5 (both in factor 2) are affected by the same predictors, and item 1 and 6 (in different factors in our model) have different pattern of significant predictors. Moreover, the results of the structural equation modeling analysis indicated that different variables are associated with the two different cyberbullying factors. 

Response to Reviewer 1

C1.1. I consider this research to be rigorous and useful to the scientific community.

However, we should improve some aspects to favor the reader's understanding. These are minor revisions.

R1.1. We are thankful for the reviewer’s appreciation of our work, and for all the suggestions for improvement that we have followed in the revision process, as detailed in the following points.

C1.2. Abstract: add all sociodemographic data, sex and age.

R1.2. We have added sex, age and geographical area information in the abstract. 

C1.3. Theoretical framework: add more meta-analysis references that support your theoretical framework. Here are some studies that could be interesting:

Pan, Y. C., Chiu, Y. C., & Lin, Y. H. (2020). Systematic review and meta-analysis of epidemiology of internet addiction. Neuroscience & Biobehavioral Reviews, 118, 612-622. https://doi.org/10.1016/j.neubiorev.2020.08.013.

Modecki, K. L., Minchin, J., Harbaugh, A. G., Guerra, N. G., & Runions, K. C. (2014). Bullying Prevalence Across Contexts: A Meta-analysis Measuring Cyber and Traditional Bullying. Journal of Adolescent Health, 55(5), 602-611. https://doi.org/10.1016/j.jadohealth.2014.06.007

Marciano, L., Schlz, P. J., & Camerini, A. L. (2020). Cyberbullying perpetration and victimization in youth: A meta-analysis of longitudinal studies. Journal of Computer-Mediated Communication, 25(2), 163-181. https://bit.ly/3iK8b0g

Lei, H., Li, S., Chiu, M. M., & Lu, M. (2018). Social support and Internet addiction among mainland Chinese teenagers and young adults: A meta-analysis. Computers in Human Behavior, 85, 200-209. https://doi.org/10.1016/j.chb.2018.03.041. https://doi.org/10.1016/j.chb.2018.03.041

Lozano-Blasco, R., Cort'es-Pascual, A., & Latorre-Martínez, M. P. (2020). Being a cybervictim and a cyberbully - The duality of cyberbullying: A meta-analysis. Computers in Human Behavior, 111, 106444. https://doi.org/10.1016/j.chb.2020.106444.

Lozano-Blasco, R.; Cortés-Pascual, A. Problematic Internet uses and depression in adolescents: A meta-Analysis. Comunicar 2020, 28, 109-120. [Google Scholar] [CrossRef].

Holt, M. K., Vivolo-Kantor, A. M., Polanin, J. R., Holland, K. M., DeGue, S., Matjasko, J. L., Wolfe, M., & Reid, G. (2015). Bullying and Suicidal Ideation and Behaviors: A Meta-Analysis. PEDIATRICS, 135(2), e496-e509. https://doi.org/10.1542/peds.2014-1864.

R1.3. We are grateful for suggestions to supplement the theoretical framework: 4 references have been included and briefly illustrated in the introductory sections of the manuscript.

C1.4. Methodology: add a main figure where the procedure and research design are visually explained.

R1.4. We have inserted a new figure (Figure 1 - below) in the revised manuscript which summarizes visually the research methodology and its main steps. We thank the Reviewer for suggesting this. 

C1.5. Participants: write the sociodemographic information in a table to make it more visual. Clarify the regions that have been involved in Italy, a heat map would bring innovation and creativity.

R1.5. We have added a table with demographic information (Table 1), including information about geographical distribution. We thank the reviewer for the suggestion of inserting a heatmap for the geographical distribution, but decided not to follow it, to reduce the number of figures and since it would have provided only limited information in addition to what is now presented in Table 1. 

C1.6. Instruments: very well developed but do not forget to put the Italian version next to the name. In the references it is fine, but it is important for the scientific community to see that you have used standardized tests adapted to their culture.

R1.6. We have specified that we used the Italian version of the scales used, and included a reference to the study that includes information about the validity of the scales. 

C1.7. Statistical analysis: very well developed, do you think that mediation analysis could provide more information on the moderating variables?

R1.7. We thank the reviewer for suggesting this analysis. We have followed the reviewer’s suggestion and conducted additional analyses to investigate mediation and moderation. We have included in the SEM model direct effects for each of the personality traits on moral disengagement as a possible mediator (of their effect on the cyberbullying factors), and formally tested the indirect effects mediated by moral disengagement (the results are discussed in the revised manuscript and the test of all the indirect and total effects are reported in a new table - Table 7). Moreover, to investigate the role of categorical factors as as gender, time on social networks and number of SN profiles as moderators of the (direct and mediated) effects of personality and moral disengagement on cyberbullying, we have conducted multigroup SEM analyses. 

C1.8. Results: very interesting but Figure 1 has poor visual quality. The concepts are very good but it looks very blurry and the size of the numbers is very small. The same happens with figure 2. Try to use some application like Canva, Flourish that allows you to improve the quality.

R1.8. We completely agree with the comment of the reviewer about the low quality of the images. However, the files that we uploaded during the submission were all high-resolution, high-quality images (600 dpi), and their quality was downgraded somehow by the submission system that created the pdf of the manuscript. We have inserted here the high-quality version of both figures (one of these figures has been changed in response to a query by the second reviewer and of new analyses conducted. If we will be allowed to submit the revised version, we will try to upload even higher-resolution files. 

C1.9. Discussion: I liked it very much, but to give it a plus of quality I recommend that you review more references on the use of social networks and cyberbullying. Add more meta-analysis studies that allow you to give more robustness to your claims. Adding the hypotheses in the text was a great idea. The limitations are very sincere and clear, but I miss two paragraphs on practical implications this research would have.

R1.9. Two new references (Pan et al., 2020; Lozano-Blasco et al., 2022) concerning meta-analysis studies on internet addiction have been added - one of which briefly discussed, to support our claims. We have also added some considerations about practical implications of our research at the end of the discussion.

C1.10. Conclusions: they are great, but you usually don't put references in this section, could you upload those references to discussion and rewrite those sentences in a more general way?

R1.10. We have revised the conclusion, moving the references to the discussion section and re-writing some sentences. 

C1.11. Supplementary material: it is very honest that you upload the questionnaire, but it is an opportunity for this to be used by others. I would encourage you to upload the data of your rating along with it, so that it can be used.

R1.11. We are glad that the reviewer appreciated the inclusion of the questionnaire in the supplementary materials. We are not sure to understand what the reviewer would like us to upload, since we already provided the ratings for the cyberbullying items. If the reviewer intended that we should also provide the ratings for the personality and moral disengagement items as well (and not only the aggregated scores), they can be already found in SM2.zip (file SM2_data_descriptives.RData). 

Response to Reviewer 2

C2.1. Despite the great efforts, authors are advised to make substantial changes to the manuscript as stated below. Because the review found some limitations regarding methods and results, comments on the discussion section are not addressed.

R2.1. We have revised the manuscript in order to address the issues raised by the reviewer, as indicated in the following points.

C2.2 Methods: Please mention the data collection period and explain the sampling method used

R2.2. We have included in the methods section information about the data collection period and sampling method. We thank the reviewer for pointing out the lack of this information in the original manuscript.

C2.3 Methods: Were the participants in the study all high school students? It seems like adolescents and young adults are mixed as authors included the age range from 14 to 25.

R2.3. Yes, the participants were all high school students. The 19 participants that were in the 21-25 age group were all 21.

C2.4 Methods: Where is the reference to the use of social networks and feelings related to network use questionnaires? Please provide the original reference and how they were modified.

R2.4. There is no reference to these items. Maybe it was not sufficiently clear in the method section that these items were devised ad hoc for this study to measure different facets of social networks use. Actually, all the questions have been reported in the text. They were not meant to provide an aggregate measure of social media use, but as ways to profile participants along some specific characteristics of their use of social networks (e.g. time), in order to investigate differences in cyberbullying across these variables, as it is common to do in similar research [e.g. reference 39]. Concerning the measure about the feelings related to social network use, we had already acknowledged among the limitations of the study the fact that this dimension was not measured with sufficient reliability. 

[38] Park S, Na E-Y, Kim E-M. The relationship between online activities, netiquette and cyberbullying. Child Youth Serv Rev. 2014; 42: 74–81. doi: 10.1016/j.childyouth.2014.04.002.

C2.5. Methods: Are the scales used in the survey all well-validated and -translated in Italian? If not validated, how was it applied to the survey? Did the researcher manually translate the items? If the scales are modified from the original version, please indicate both the original and revised version.

R2.5. The scale used for measuring personality was the validated Italian version of the 10 item Personality Inventory [ref 70]. The scales used to measure cyberbullying and moral disengagement about cyberbullying were validated in English [ref 51], and had been translated in Italian by the authors for a previous study, in which Confirmatory Factor Analysis and SEM validated the translation replicating the findings of the study that introduced these scales. [ref 68]

[70] Guido G, Peluso AM, Capestro M, Miglietta M. An Italian version of the 10-item Big Five Inventory: An application to hedonic and utilitarian shopping values. Pers Individ Differ. 2015; 76: 135–140. doi:. 10.1016/j.paid.2014.11.053.

[51] Meter DJ, Bauman S. Moral disengagement about cyberbullying and parental monitoring: Effects on traditional bullying and victimization via cyberbullying involvement. J Early Adolesc. 2018. 38: 303–326. doi: 10.1177/0272431616670752. 

[68] Parlangeli O, Marchigiani E, Guidi S, Bracci M, Andreadis A, Zambon R. I Do It Because I Feel that...Moral Disengagement and Emotions in Cyberbullying and Cybervictimisation. In: Meiselwitz G., editor. Social Computing and Social Media. Design, Ethics, User Behavior, and Social Network Analysis. HCII 2020. Lecture Notes in Computer Science, vol 12194. Cham: Springer; 2020. pp. 289-304. doi: 10.1007/978-3-030-49570-1_20.

C2.6. Results: It seems like the results of chi-square is only reported in the text, did the authors not create table?

R2.6. We did create the tables for all the chi-square tests conducted, but we omitted them from the manuscript choosing to highlight the main findings (for which groups frequencies differed). (Note: it is not clear from the reviewer’s comment which tables the reviewer had in mind). We could include these tables as supplementary materials of the revised manuscript.

C2.7. Results: Authors attempted to test several hypotheses but having so many hypotheses often mislead the readers to lose focus of the study. To make the results section more reader-friendly, it is recommended for the authors to match the results with the corresponding hypotheses so that they/we could understand why these results are presented in this section.

R2.7. Although we understand that the many results presented in the results section, deriving from a complex study addressing several research questions and testing several hypothesis, might mislead the readers to lose focus of the study, we also believe that the way the results are currently organized is logical and follow the logical order highlighted in the statistical analysis section in order to answer the research questions. Moreover, the discussion section (not addressed by the comments by the reviewer) matches the results to the research questions and hypotheses. We have however edited the results subsections titles in order to make it more evident the link with the research questions and hypotheses that the results in each subsection are functional to address. Also, in response to a suggestion by Reviewer 1 we have included immediately before the Results section a visual summary of the methodology of the study, in which the research questions are put in correspondence with the different types of data analyses conducted. 

C2.8. Results: When reviewing the methods section, I thought six items used to measure cyberbullying would be combined as a single variable instead of splitting each item. If the purpose of the current study was to measure the different types of cyberbullying, it was more appropriate to choose a scale consisting of several subscales rather than splitting a single measurement tool by items. This is a big limitation therefore it should be clearly stated in the limitation section.

R2.8. Actually, this was the aim of the study: to verify whether a scale, made up of “only” 6 items, and which is usually used as a tool to measure cyberbullying as a unitary phenomenon, could not instead be revealing of different aspects, different reasons and different manifestations of harmful virtual behavior. To test this hypothesis we tried to relate the personality traits and the mechanisms of Moral Disengagement with each of the items, and again for this reason we conducted the factor analysis.

The results go in the hypothesized direction: for each of the behaviors related to each item, different relationships with personality traits can be highlighted. Factor analysis also suggested the existence of at least two factors.

We hope that in the revised version of our manuscript this will be clearer.

C2.9. Results: I don’t quite understand the intention behind conducting the EFA, CFA, and structural equation modelling in this study. This study did not develop a scale nor did it validate the existing scale. Why should these analyses be performed? For what purpose?

R2.9. The reasons for conducting the analyses highlighted in the reviewer’s comment are first of all related to research question 4, in conjunction with the previous research questions. This research question concerned whether cyberbullying is better conceived and measured as a unitary construct or as a multidimensional construct, and to this aim exploratory factor analysis followed by confirmatory factor analysis is a common analytical strategy. In a way, it can be seen as an attempt to validate the scale by Meter and Bauman, since the items that we used to measure cyberbullying were taken from that scale. The SEM analysis was conducted to test hypotheses about the association between the different latent factors of cyberbullying, personality traits, moral disengagement and other variables. In the revised version of the manuscript we present additional multigroup SEM analyses conducted to better investigate differences across groups and to test hypotheses about mediation (by moral disengagement) and moderation (by gender, time on social networks and number of profiles) on the effect of personality and moral disengagement on the different types of cyberbullying (as measured by the two latent factors extracted by the previous steps - EFA/CFA). We have revised the statistical analysis and results sections of the paper to make it clearer what was the purpose of all these analyses. 

C2.10. Results: Figure 2 is very confusing, and it is not a typical arrangement for SEM. Please refer back to other SEM research to redraw the figure. Why is there no latent variable for openness, extraversion etc. and for social network uses? Just by looking at this figure, the DV and IV are unclear.

R2.10. We realize that figure 2 was not a typical display of a SEM model, as we had introduced some mapping between coefficients’ size/sign/significance and visual features (width, color and linetype) to make some patterns more evident. But in the revised manuscript, that includes the results of new analyses conducted in response to a comment by Reviewer 1, we have redrawn the figure (now Figure 3) making it more similar to typical SEM path diagrams. However, we have kept the color coding scheme for the paths representing the direction of the association, and we have also used dashed lines to represent insignificant path coefficients in the fitted model, as we believe it makes it easier to read the effects in complex models like this one. 

C2.11: Lastly, the sentences are way too long in some paragraphs. Please polish the language in order to improve the overall quality of the manuscript.

R2.11. We have edited the language to improve readability and overall manuscript quality. 

Response to Reviewer 3

C3.1. This paper conducted a questionnaire that was administered to a large sample of high school students to validate the four hypotheses regarding cyberbullying. These research questions are interesting and provide insights about cyberbullying behavior. For example, statistical analyses in this work show that cyberbullying is not a unitary construct but a multidimensional construct.

R3.1. We thank reviewer 3 for appreciating the underlying rationale of our study and the results we obtained

C3.2. I have following suggestions to further improve the manuscript:

1. Given the extreme popularity of AI and Machine Learning, in related work, the authors might consider including recent works that used machine learning models to validate similar findings. For example, in [1], the authors showed that user's personality traits and peer influence are important predictors of cyberbullying. In [2], the authors also showed capturing the repetitive pattern in cyberbullying behavior can improve the performance.

[1] PI-bully: Personalized cyberbullying detection with peer influence

[2]Hierarchical attention networks for cyberbullying detection on the instagram social network

R3.2. In fact, in the previous version of this manuscript there was only one reference related to this important field of research.

Now, in the introduction, 7 more references have been added and discussed, including those suggested by the reviewer.

C3.3. 2. The figures are very low-quality and it is hard to read. The authors need to replace them with high-resolution figures.

R3.3. As we stated in R1.X We agree on the low quality of the figures. However, the files that we uploaded during the submission were all high-resolution, high-quality images (600 dpi), and their quality was downgraded somehow by the submission system that created the pdf of the manuscript. We are sure that in the final version of this manuscript all figures will be high-quality images.

C3.4. 3. The cyberbullying definition is not clear, and sometimes confuses with other similar concept, such as cyber-aggression [3]. The authors need to clarify these differences.

[3]Cyberbullying: Bullying in the digital age

R3.4. In the introduction it has been clarified, also reporting two new references, which is the generally shared definition of cyberbullying and its differences with cyber incivility and cyber aggression

---

## [Decision Letter · Decision Letter 1]

10 May 2022

How many cyberbullying(s)? A non-unitary perspective for offensive online behaviours

PONE-D-21-37456R1

Dear Dr. Guidi,

We’re pleased to inform you that your manuscript has been judged scientifically suitable for publication and will be formally accepted for publication once it meets all outstanding technical requirements.

Kind regards,

Sergio A. Useche, Ph.D.

Academic Editor

PLOS ONE

3. Please remove your figures from within your manuscript file, leaving only the individual TIFF/EPS image files, uploaded separately. These will be automatically included in the reviewers’ PDF.

4. Please ensure that you refer to Figure 1 in your text as, if accepted, production will need this reference to link the reader to the figure.

Additional Editor Comments (optional):

Reviewers' comments:

Reviewer's Responses to Questions

**Comments to the Author**

1. If the authors have adequately addressed your comments raised in a previous round of review and you feel that this manuscript is now acceptable for publication, you may indicate that here to bypass the “Comments to the Author” section, enter your conflict of interest statement in the “Confidential to Editor” section, and submit your "Accept" recommendation.

Reviewer #1: All comments have been addressed

2. Is the manuscript technically sound, and do the data support the conclusions?

Reviewer #1: Yes

3. Has the statistical analysis been performed appropriately and rigorously? 

Reviewer #1: (No Response)

4. Have the authors made all data underlying the findings in their manuscript fully available?

Reviewer #1: Yes

5. Is the manuscript presented in an intelligible fashion and written in standard English?

Reviewer #1: Yes

6. Review Comments to the Author

Reviewer #1: Dear Editor and Authors,

The research is of high quality, the authors have greatly improved their work. In addition, they have been able to respond to all the reviewers' responses in a consistent manner. I would also like to emphasize the value of this research as it brings very important and interesting information to the use of new technologies. My sincere congratulations.

7. PLOS authors have the option to publish the peer review history of their article (what does this mean?). If published, this will include your full peer review and any attached files.

Reviewer #1: **Yes: **

---

## [Editor Report · Acceptance letter]

30 May 2022

PONE-D-21-37456R1 

How many cyberbullying(s)? A non-unitary perspective for offensive online behaviours 

Dear Dr. Guidi:

I'm pleased to inform you that your manuscript has been deemed suitable for publication in PLOS ONE. Congratulations! Your manuscript is now with our production department. 

Kind regards, 

on behalf of

Dr. Sergio A. Useche 

Academic Editor

PLOS ONE